# Improving Diffusion Models for Class-imbalanced Training Data via Capacity Manipulation

**Feng Hong**[1,*] **Jiangchao Yao**[1,✉] **Yifei Shen**[2]
**Dongsheng Li**[2] **Ya Zhang**[3,4] **Yanfeng Wang**[3,✉]

[1]Cooperative Medianet Innovation Center, Shanghai Jiao Tong University
[2]Microsoft Research Asia
[3]School of Artificial Intelligence, Shanghai Jiao Tong University
[4]Institute of Artificial Intelligence for Medicine, Shanghai Jiao Tong University School of Medicine

```
{feng.hong, Sunarker, ya_zhang, wangyanfeng}@sjtu.edu.cn
{yifeishen, dongsli}@microsoft.com
```

## Abstract

While diffusion models have achieved remarkable performance in image generation, they often struggle with the imbalanced datasets frequently encountered in real-world applications, resulting in significant performance degradation on minority classes. In this paper, we identify model capacity allocation as a key and previously underexplored factor contributing to this issue, providing a perspective that is orthogonal to existing research. Our empirical experiments and theoretical analysis reveal that majority classes monopolize an unnecessarily large portion of the model's capacity, thereby restricting the representation of minority classes. To address this, we propose Capacity Manipulation (CM), which explicitly reserves model capacity for minority classes. Our approach leverages a low-rank decomposition of model parameters and introduces a capacity manipulation loss to allocate appropriate capacity for capturing minority knowledge, thus enhancing minority class representation. Extensive experiments demonstrate that CM consistently and significantly improves the robustness of diffusion models on imbalanced datasets, and when combined with existing methods, further boosts overall performance. [1]

## 1 Introduction

Diffusion models have demonstrated exceptional generative capabilities, as evidenced by their rapid adoption in both academia and industry (Ho et al., 2020; Song et al., 2021b; Dhariwal and Nichol, 2021; Ramesh et al., 2022; Rombach et al., 2022; Peng et al., 2026). Nevertheless, a key challenge persists: diffusion models exhibit a marked decline in performance on minority classes when trained on class-imbalanced datasets (Qin et al., 2023; Zhang et al., 2024), a concern amplified by the prevalence of data imbalance in real-world scenarios (Reed, 2001; Zhang et al., 2023).

Most research on imbalanced learning has focused on enhancing the robustness of discriminative models (Buda et al., 2018; He and Garcia, 2009; Wang et al., 2021a; Menon et al., 2021a; Cui et al., 2021). However, these techniques cannot be directly applied to diffusion models due to fundamental differences in model architectures, training, and inference processes. To address class imbalance in generative diffusion models, recent pioneering works (Qin et al., 2023; Zhang et al., 2024) have focused on loss function design to facilitate knowledge transfer between majority and minority classes. For example, Class Balancing Diffusion Models (CBDM) (Qin et al., 2023) introduced a regularization loss that implicitly encourages generated images to follow a balanced prior distribution at each sampling step. Oriented Calibration (OC) (Zhang et al., 2024) further improved the generation quality of minority classes through knowledge transfer from majorities.

---

[*] Work partially done during Feng Hong's internship at Microsoft Research Asia.
[1]Code: https://github.com/Feng-Hong/ImbDiff-CM.

In this paper, we explore an orthogonal perspective: *model capacity allocation*[2]. Our motivation stems from the empirical observation that majority classes tend to monopolize a disproportionate share of the model's capacity, thus limiting the capacity available for minority classes. This finding is supported by both empirical and theoretical analyses. Empirically, the visual quality of minority classes is significantly lower than majority classes. Furthermore, we find that although the training loss for minority classes is similar to majority classes, pruning model parameters with the smallest L1-Norm leads to a more pronounced change in relative loss for minority classes. This suggests that minorities are allocated less model capacity, making them particularly vulnerable to pruning. A gradient-based theoretical analysis further supports this, showing that the majority classes drive a significant fraction of parameter updates, thereby dominating capacity allocation.

To address this issue, we propose a dedicated model capacity allocation method for diffusion models, termed Capacity Manipulation (CM). Our core idea is to reserve model capacity specifically for minority expertise in advance, preventing it from being overtaken by majority classes and thus safeguarding the learning process for minority samples. Concretely, we decompose model parameters into two components using low-rank techniques: one capturing majority and general knowledge, and the other reserved for minority expertise (Eq. (1)). By introducing a capacity manipulation loss (Eq. (2)), we ensure that minority knowledge is effectively allocated its reserved capacity during training. We further provide theoretical support, proving that our method can successfully balance update proportions to ensure adequate capacity allocation for minority classes. Our key contributions are summarized as follows:

- **A Novel Perspective:** Unlike prior works, this paper takes the first step in exploring model capacity allocation to enhance diffusion models on class-imbalanced training data, motivated by both empirical findings and theoretical analysis.

- **A New Method:** We propose a new method, CM, which protects minority classes by reserving model capacity for minority expertise and strategically allocating relevant knowledge during training. Instead of increasing model size, CM reallocates existing capacity, introducing no additional inference overhead. Furthermore, CM is orthogonal to existing methods and can be integrated complementarily.

- **Excellent Performance:** Extensive experiments on various datasets and diffusion model architectures demonstrate that our method consistently outperforms state-of-the-art baselines. CM substantially improves minority class performance without sacrificing generative quality on majority classes.

## 2 ON THE LIMITATIONS OF DIFFUSION MODELS ON IMBALANCED DATA

### 2.1 PROBLEM FORMULATION

Let $\mathcal{X}$ be the image space and $\mathcal{Y} = \{1, \ldots, C\}$ the label space, where $C$ is the class number. We consider an imbalanced training set $\mathcal{D} = \{(\mathbf{x}^n, y^n)\}_{n=1}^N \in (\mathcal{X}, \mathcal{Y})^N$ of size $N$. The per-class sample counts $N_c$ for $c \in \mathcal{Y}$ in the descending order exhibit a long-tailed distribution. The goal is to learn a generative diffusion model $p_\theta(\mathbf{x}|y)$, parameterized by $\theta$, from the imbalanced data $\mathcal{D}$ to generate realistic and diverse samples across all classes. For unconditional generation, the class condition can be set to Null, yielding $p_\theta(\mathbf{x}) = p_\theta(\mathbf{x}|\text{Null})$.

### 2.2 EMPIRICAL MOTIVATIONS

Diffusion models, despite their strengths, are notably vulnerable to data imbalance. We illustrate this in Fig. 1(a) with a DDPM trained on Imb. CelebA-HQ (100:1 female-to-male sample ratio). While good at generating majority-class (female) images, its performance on minority-class (male) images is severely degraded, evident from visual results and FID scores. In comparison, we illustrate a preview of our proposed method, which significantly improves the generation quality for the minority class (male images). Fig. 2(c) demonstrates similar findings on Imb. CIFAR-100.

---

[2]In this paper, we define "model capacity" as the *available representational resources* of the neural network to capture and store data features.

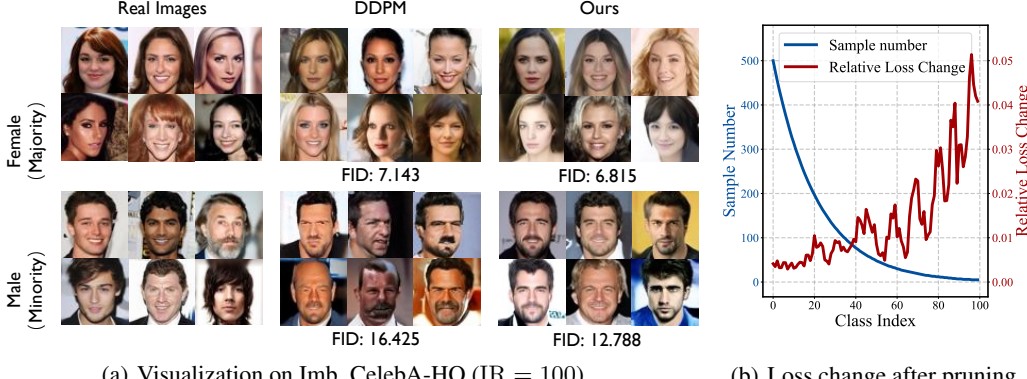

(a) Visualization on Imb. CelebA-HQ ($\mathrm{IR} = 100$)    (b) Loss change after pruning

Figure 1: (a) Real images from Imb. CelebA (Imbalance Ratio $\mathrm{IR} = 100$) and generated images from DDPM and our method. Images are randomly sampled. (b) Relative per-class loss change, defined as $(\mathcal{L}^c_{pruned} - \mathcal{L}^c_{raw})/\mathcal{L}^c_{raw}$, for a DDPM model trained on Imb. CIFAR-100 ($\mathrm{IR} = 100$) when 10% of its parameters with the smallest L1-Norm are pruned. Raw losses and absolute loss changes are provided in Figs. G.1 and G.2.

Fig. 1(b) reveals an intriguing finding on Imb. CIFAR-100: despite similar raw training losses across classes (Fig. G.1), minority classes are disproportionately sensitive to pruning the smallest L1-norm parameters. A model capacity perspective may offer a compelling explanation: majority classes monopolize capacity, forcing minority class information into less critical (low L1-norm) parameters. These fragile representations are thus more susceptible to pruning, underscoring standard diffusion model limitations and motivating targeted capacity allocation.

## 2.3 THEORETICAL INSIGHTS

We provide Theorem 2.1 to serve as an illustrative example: majority classes dominate the model's capacity, leaving fewer resources for minorities, which leads to poorer performance for them. The assumption, detailed proof, and its **extension to the multi-class setting** are presented in Sec. F.

**Theorem 2.1.** *Consider a setting with two classes $\mathcal{Y} = \{1, 2\}$, where the training set is highly imbalanced, i.e., $N_1 \gg N_2$. Under Assumption F.1, the expected proportion of parameter matrices within the network whose update direction are dominated by the majority class $y = 1$ is given by*
$$\Pi_{maj} = \Phi\left(\frac{(2a-1)\mu\sqrt{2N(1-\cos\angle(\mu_1,\mu_2))}}{2\sigma}\right),\ \text{where } a = \frac{N_1}{N_1+N_2} \text{ is the majority ratio, } N = N_1 + N_2,$$
$\Phi(\cdot)$ *is the standard normal cumulative distribution function, $\mu_1$ and $\mu_2$ are expected gradients of classes $y = 1$ and $y = 2$, $\mu > 0$ and $\sigma > 0$ are positive constants defined in Assumption F.1.*

**Remarks.** We can draw the following insight from Theorem 2.1:

- When $a > 0.5$ and approaches 1, the standard normal CDF $\Phi(\cdot)$ (monotonic, concave, and $> 0.5$ for positive inputs, Eq. (F.6)) implies: (1) the majority class dominates model capacity, and (2) greater class imbalance leads to greater capacity imbalance, albeit with diminishing returns.
- $\Pi_{\mathrm{maj}}$ decreases as inter-class similarity increases (smaller $\angle(\mu_1, \mu_2)$), suggesting greater class similarity allows minority classes to benefit more from majority-driven updates.
- $\Pi_{\mathrm{maj}}$ decreases with relative within-class diversity ($\frac{\sigma}{\mu}$). Higher diversity helps mitigate imbalance.
- $\Pi_{\mathrm{maj}}$ increases with total samples $N$, suggesting capacity dominance worsens for larger datasets.

## 3 METHOD: CAPACITY MANIPULATION

Based on the above motivations, we propose allocating dedicated model capacity specifically for minority classes, reserving it in advance to prevent encroachment by majority classes

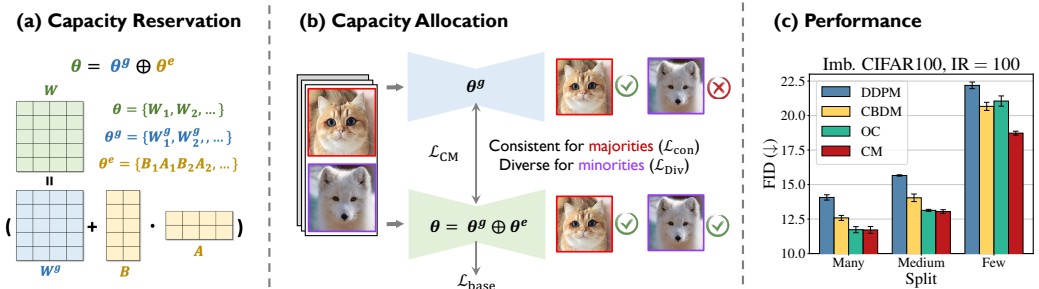

Figure 2: (a,b) An overview of our method, CM. (a) An illustration of the capacity reservation part of CM. (b) An illustration of how CM allocates the corresponding knowledge to the reserved model capacity during training. (c) Many/Medium/Few split performance on Imb. CIFAR100 with imbalance ratio $\text{IR} = 100$, where Many/Medium/Few represents the top, middle, and bottom thirds of classes sorted by sample number in descending order. CM significantly improves minority performance without sacrificing the performance of majorities.

## 3.1 CAPACITY RESERVATION

To allocate sufficient model capacity for minorities, we first need to explicitly partition the model capacity. Here we achieve this purpose by a technique similar to Low-Rank Adaptations (LoRAs) (Hu et al., 2022), which has demonstrated excellent performance and versatility in the field of efficient fine-tuning. While our task and goal differ, we apply its low-rank decomposition concept to partition the model capacity. For a diffusion model parameterized by $\theta = \{W_1, W_2, \ldots\}$, where each $W \in \theta$ represents a parameter matrix in the network, we decompose any $W \in \mathbb{R}^{d \times k}$ as

$$W = W^g + BA = W^g + W^e, \forall W \in \theta, \tag{1}$$

where $W^g \in \mathbb{R}^{d \times k}$ represents the part of $W$ to be retained for majorities and generalized knowledge, $W^e = BA \in \mathbb{R}^{d \times k}$ represents the part to be allocated to the minority expertise, $B \in \mathbb{R}^{d \times r}$, $A \in \mathbb{R}^{r \times k}$, and the rank $r < \min(d, k)$. From Eq. (1), we decompose $\theta$ into $\theta^g = \{W_1^g, W_2^g, \ldots\}$ and $\theta^e = \{W_1^e, W_2^e, \ldots\}$ and merge them by $\theta = \theta^g \oplus \theta^e$, where $\oplus$ means the element-wise addition. An illustration of Capacity Reservation is shown in Fig. 2(a).

As a follow-up to Theorem 2.1, we present Theorem 3.1, which demonstrates how the low-rank parameter decomposition in Eq. (1) for reserving capacity affects the actual capacity among classes.

**Theorem 3.1.** *For two classes $\mathcal{Y} = \{1, 2\}$, assume the training set is highly imbalanced, i.e., $N_1 \gg N_2$. Suppose that class $y = 2$ contributes gradients to both $W^g$ and $W^e$, while class $y = 1$ only contributes to $W^g$. Under Assumptions F.1 and F.2, the expected proportion of parameter matrices $W \in \theta$ within the network whose update direction are dominated by the majority class $y = 1$ is bounded by $\Pi_{maj} < \Phi\left(\frac{(1-\alpha_r)\mu\sqrt{2N(1-\cos\angle(\mu_1,\mu_2))}}{2\sigma}\right)$, where $\alpha_r$ is a monotonically increasing function of rank $r$, with $\alpha_r = 0$ when $r = 0$, and $\alpha_r = 1$ when $W^e$ is full rank.*

In this bound, we omit the effect of the inter-class sample ratio by manipulating the inequality, in order to clearly showcase the impact of the low-rank component. Consequently, the rank of $W^e$, modulated by $\alpha_r$, controls the capacity allocation between majority and minority classes. Adjusting the rank directly mitigates the capacity collapse inherent in standard training. We provide the assumptions and detailed proof of Theorem 3.1 in Sec. F. We also provide extensions of Theorem 2.1 and Theorem 3.1 to the multi-class setting in Sec. F.4.

## 3.2 CAPACITY ALLOCATION

With the model parameters decomposed as $\theta = \theta^g \oplus \theta^e$, our goal during training is to allocate $\theta^e$ for minority expertise and $\theta^g$ for majority and generalized knowledge, enhancing minorities through capacity manipulation. To achieve this, the diffusion model $p_\theta(x|y) = p_{\theta^g \oplus \theta^e}(x|y)$ should perform well on all samples, both majorities and minorities. Meanwhile, $p_{\theta^g}(x|y)$ should perform well on majorities but poorly on minorities, as $\theta^g$ is not intended to learn the minority expertise.

**Capacity manipulation loss.** For $\theta = \theta^g \oplus \theta^e$, we use base loss $\mathcal{L}_{\text{base}}(\mathcal{D}, \theta)$ to train on the entire dataset, covering both majorities and minorities. Here, we can use the commonly adopted diffusion loss as the base loss: $\mathcal{L}_{\text{base}}(\mathcal{D}, \theta) = \frac{1}{N} \sum_{x, y \in \mathcal{D}} \mathbb{E}_{\epsilon, t} \left[ \| \epsilon_\theta(x_t, t, y) - \epsilon \|_2^2 \right]$. Alternatively, we can choose variants adapted for imbalanced datasets, *e.g.*, Zhang et al. (2024); Qin et al. (2023). For the separate $\theta^g$ and $\theta^e$, we propose a capacity manipulation loss, which encourages $\theta^e$ to learn minority expertise and $\theta^g$ to capture generalized knowledge:

$$\mathcal{L}_{\text{CM}}(\mathbf{x}, y, \theta^g, \theta^e) = \mathcal{L}_{\text{Con}}(\mathbf{x}, y, \theta^g, \theta^e) + \mathcal{L}_{\text{Div}}(\mathbf{x}, y, \theta^g, \theta^e),$$
$$\mathcal{L}_{\text{Con}}(\mathbf{x}, y, \theta^g, \theta^e) = \omega_{\text{Con}}^y \mathbb{E}_t \| \epsilon_{\theta^g \oplus \theta^e}(\mathbf{x}_t, t, y) - \epsilon_{\theta^g}(\mathbf{x}_t, t, y) \|_2^2, \quad (2)$$
$$\mathcal{L}_{\text{Div}}(\mathbf{x}, y, \theta^g, \theta^e) = -\omega_{\text{Div}}^y \mathbb{E}_t \| \epsilon_{\theta^g \oplus \theta^e}(\mathbf{x}_t, t, y) - \epsilon_{\theta^g}(\mathbf{x}_t, t, y) \|_2^2,$$

where the capacity manipulation loss $g\mathcal{L}_{\text{CM}}$ is composed of a consistency loss $\mathcal{L}_{\text{Con}}$ and a diversity loss $\mathcal{L}_{\text{Div}}$. We vary the consistency class weight $\omega_{\text{Con}}$ and the diversity class weight $\omega_{\text{Div}}$ applied to different classes. For class $c \in \mathcal{Y}$ with $N_c$ instances, a larger $N_c$ (majorities) results in a higher consistency class weight $\omega_{\text{Con}}^c$, leading to more consistent outputs between $\epsilon_{\theta^g \oplus \theta^e}(\mathbf{x}_t, t, y)$ and $\epsilon_{\theta^g}(\mathbf{x}_t, t, y)$. Conversely, for the diversity class weight, a smaller $N_c$ (minorities) results in a higher $\omega_{\text{Div}}^c$, leading to more diverse outputs between $\epsilon_{\theta^g \oplus \theta^e}(\mathbf{x}_t, t, y)$ and $\epsilon_{\theta^g}(\mathbf{x}_t, t, y)$. Thus, $p_{\theta^g}(x|y)$ excels on majorities, as its output aligns with $\theta$, but underperforms on minorities due to the divergence between the outputs of $\theta^g$ and $\theta$. Specifically,

$$\omega_{\text{Con}}^y = \frac{C N_y}{\sum_{c=1}^C N_c}, \quad \omega_{\text{Div}}^y = \frac{C}{N_y \sum_{c=1}^C \frac{1}{N_c}}. \quad (3)$$

Here $\omega_{\text{Con}}$ scales linearly with class sample size, while $\omega_{\text{Div}}$ is inversely proportional to class sample size, ensuring $\omega_{\text{Con}} = \omega_{\text{Div}} = 1$, $\mathcal{L}_{\text{CM}}(\mathbf{x}, y, \theta^g, \theta^e) = 0$ for a balanced dataset. $\omega_{\text{Con}}$ and $\omega_{\text{Div}}$ adaptively assign smooth weights to classes with varying sample sizes, avoiding the need for manual separation between majorities and minorities. *While formally similar to reweighting techniques in discriminative models (Cui et al., 2019b) and generative models (Xie et al., 2023; Fan et al., 2024; Kim et al., 2024; Li et al., 2025; Liu et al., 2025), Eq. (3) differs fundamentally: it allocates model capacity by modulating output distances tied to distinct network parameters, rather than weighting individual sample losses.*

**Joint optimization.** For $\theta = \theta^g \oplus \theta^e$, we optimize the base loss $\mathcal{L}_{\text{base}}$ and the capacity manipulation loss $\mathcal{L}_{\text{CM}}$, weighted by hyperparameter $\lambda$:

$$\min_\theta \mathcal{L}_{\text{Total}}(\mathcal{D}, \theta) = \mathcal{L}_{\text{base}}(\mathcal{D}, \theta) + \lambda \sum_{(\mathbf{x}, y) \in \mathcal{D}} \frac{1}{N} \mathcal{L}_{\text{CM}}(\mathbf{x}, y, \theta^g, \theta^e), \quad (4)$$

where $\mathcal{L}_{\text{base}}$ optimizes $\theta$ for both majorities and minorities, while the capacity manipulation loss $\mathcal{L}_{\text{CM}}$ acts as a regularizer to allocate capacity and protect minorities. This guides $\theta$ toward more balanced and effective model weights. An illustration of the training process is presented in Fig. 2(b).

**Inference.** For inference, we can explicitly compute and store $\theta = \theta^g \oplus \theta^e$, and sample images from $p_\theta(\mathbf{x}|y)$. Thus, our method *does not increase model capacity*, ensuring *no additional inference latency* compared to a standard model, which is crucial as inference speed is a key bottleneck in real-world deployment (Song et al., 2021a). This advantage comes from the LoRA-like parameter decomposition in Eq. (1) and explicitly aggregating parameters during inference.

### 3.3 DISSCUSSION

**Comparison with existing imbalanced diffusion models.** Unlike current methods such as CBDM and OC, which focus on designing improved objective functions for imbalanced data, CM improves the robustness of diffusion models to class imbalance from a new perspective: allocating model capacity to protect minorities. CM is orthogonal and can benefit from improved objective functions to achieve further enhancements.

**Comparison with LoRA and MoE-style Adapters.** While CM shares a low-rank structure with LoRA, our goal is capacity reservation prior to training rather than efficient fine-tuning. In Tab. 1, we compare two variants of the CBDM based on low-rank components: (1) CBDM ($\theta = \theta_g \oplus \theta_e$): This variant utilizes our decomposition approach while maintaining the original CBDM objective; (2) CBDM (LoRA): This variant incrementally fine-tunes the pre-trained CBDM model using LoRA.

Table 1: FID$\downarrow$ comparison with baselines and low-rank variants.

| Dataset | DDPM | CBDM | CBDM ($\theta = \theta^g \oplus \theta^e$) | CBDM (LoRA) | CM |
|---|---|---|---|---|---|
| CIFAR-100, $\mathrm{IR} = 100$ | 10.163 | 10.051 | 10.231 | 11.424 | **7.519** |
| CIFAR-10, $\mathrm{IR} = 100$ | 10.697 | 8.233 | 8.316 | 9.725 | **7.727** |

Neither variant yields improvements over the original CBDM, whereas CM significantly outperforms all baselines. This is because low-rank parameterizations like LoRA do not increase model capacity, but merely change how parameter updates are applied to a fixed-size network. To further distinguish CM from Mixture-of-Experts (MoE) paradigms, we implemented a "Group-Expert LoRA" baseline (details in Sec. E.4), which assigns distinct LoRA adapters to Many/Medium/Few class groups as a deterministic MoE. As shown in Sec. E.4, CM (FID 7.519) significantly outperforms this MoE-style baseline (FID 10.06 on Imb. CIFAR-100), confirming that our gains stem from explicit capacity manipulation rather than simply increasing parameters or dynamic routing. Detailed conceptual comparisons with reweighting and adapters are provided in Sec. E.3.

**Extension to LoRA-finetuning.** Our method can be seamlessly extended to LoRA-finetuning scenarios by modifying Eq. (1) to the form: $W = W^f + B^g A^g + B^e A^e$. Here, $\theta^f = \{W_1^f, W_2^f, \ldots\}$ represents the frozen pre-trained model parameters, $\theta^g = \{B_1^g A_1^g, B_2^g A_2^g, \ldots\}$ denotes the trainable parameters allocated for majorities and generalized knowledge, and $\theta^e = \{B_1^e A_1^e, B_2^e A_2^e, \ldots\}$ corresponds to the trainable parameters reserved for minority expertise. For $W \in \mathbb{R}^{d \times k}$, $B^g \in \mathbb{R}^{d \times r^g}$, $A^g \in \mathbb{R}^{r^g \times k}$, $B^e \in \mathbb{R}^{d \times r^e}$, $A^e \in \mathbb{R}^{r^e \times k}$, we have $r^e < r^g < \min(d, k)$. During inference, the model parameters are merged by $\theta = \theta^f \oplus \theta^g \oplus \theta^e$. This extension preserves the structure of LoRA while enhancing the fine-tuning process by capacity manipulation.

## 4 EXPERIMENTS

### 4.1 EXPERIMENTAL SETUP

**Datasets.** We conduct experiments on the imbalanced versions of commonly used datasets in the field of image synthesis, including CIFAR-10 (Krizhevsky et al., 2009), CIFAR-100 (Krizhevsky et al., 2009), CelebA-HQ (Karras et al., 2018), ImageNet-LT (Liu et al., 2019), iNaturalist (Horn et al., 2018), and ArtBench-10 (Liao et al., 2022). CIFAR-10 and CIFAR-100 have a resolution of 32×32; for CelebA-HQ, we use the 64×64 version; for ImageNet-LT and iNaturalist, we use the 32×32 and 64×64 versions; and for ArtBench-10, we use the original resolution of 256×256. We follow Cao et al. (2019a) to construct imbalanced versions of these datasets by downsampling, resulting in an exponential decrease in the sample size of each class with its index. We refer to these imbalanced datasets as Imb. $\mathrm{dataset}$, $e.g.$,, Imb. CIFAR-10. We control the level of imbalance in the dataset by setting different imbalance ratios (Menon et al., 2021a; Qin et al., 2023) $\mathrm{IR} \in \{50, 100\}$, where IR is the ratio of the number of samples in the most populous class to that in the least populous class, defined as $\mathrm{IR} = \frac{\max_{c \in \mathcal{Y}} N_c}{\min_{c \in \mathcal{Y}} N_c}$. For Imb. CIFAR-10 and Imb. ArtBench-10, we divide the dataset into three splits: *Many* (classes 0-2), *Medium* (classes 3-5), and *Few* (classes 6-9) based on class sizes in descending order. Similarly, for Imb. CIFAR-100, the splits are *Many* (classes 0-32), *Medium* (classes 33-65), and *Few* (classes 66-99). This uniform three-way split follows Jiang et al. (2021).

**Baselines.** We consider baselines including: (1) the base denoising diffusion probabilistic model (DDPM); (2) methods specifically targeting imbalanced diffusion models: the class-balancing diffusion model (CBDM) (Qin et al., 2023) and Oriented Calibration (OC) (Zhang et al., 2024); (3) applying imbalance learning methods from discriminative models or generative adversarial networks (GANs) to diffusion models: re-sampling (RS) (Mahajan et al., 2018), adaptive discriminator augmentation (ADA) (Karras et al., 2020), and focal loss (Lin et al., 2017b). Note that many imbalanced learning methods for discriminative models and GANs heavily rely on specific model architectures or training paradigms, $e.g.,$ Menon et al. (2021a); Zhou et al. (2023a); Rangwani et al. (2022), making them incompatible with imbalanced diffusion models.

**Metrics.** The performance is evaluated using the metrics Frechet Inception Distance (FID) (Heusel et al., 2017), Kernel Inception Distance (KID) (Binkowski et al., 2018), Recall (Kynkäänniemi et al., 2019), and Inception Score (IS) (Salimans et al., 2016). All metrics are calculated based on features

Table 2: FIDs ($\downarrow$), KIDs ($\downarrow$), Recalls ($\uparrow$), and ISs ($\uparrow$) of CM and various baseline methods on Imb. CIFAR-10 and Imb. CIFAR-100. Results for imbalance ratios IR $= \{100, 50\}$ are shown side-by-side. **Best** and second-best results are highlighted. *Results with Mean$\pm$Std can be found in Tab. G.1.*

| Imb. CIFAR-10 | IR $= 100$ | | | | IR $= 50$ | | | |
|---|---|---|---|---|---|---|---|---|
| Method | FID $\downarrow$ | KID $\downarrow$ | Recall $\uparrow$ | IS $\uparrow$ | FID $\downarrow$ | KID $\downarrow$ | Rec $\uparrow$ | IS $\uparrow$ |
| DDPM (Ho et al., 2020) | 10.697 | 0.0035 | 0.47 | 9.39 | 10.216 | 0.0035 | 0.47 | 9.37 |
| +ADA (Karras et al., 2020) | 9.266 | 0.0029 | 0.49 | 9.26 | 9.132 | 0.0030 | 0.51 | 9.28 |
| +RS (Mahajan et al., 2018) | 12.332 | 0.0037 | 0.45 | 9.25 | 11.231 | 0.0038 | 0.47 | 9.31 |
| +Focal (Lin et al., 2017b) | 10.842 | 0.0034 | 0.46 | 9.42 | 10.315 | 0.0034 | 0.48 | 9.38 |
| CBDM (Qin et al., 2023) | 8.233 | 0.0026 | **0.53** | 9.23 | 7.933 | 0.0026 | **0.54** | 9.42 |
| OC (Zhang et al., 2024) | 8.390 | 0.0027 | 0.52 | **9.53** | 8.034 | 0.0027 | 0.53 | 9.65 |
| CM | **7.727** | **0.0023** | **0.53** | 9.52 | **7.372** | **0.0024** | **0.54** | **9.69** |

| Imb. CIFAR-100 | IR $= 100$ | | | | IR $= 50$ | | | |
|---|---|---|---|---|---|---|---|---|
| Method | FID $\downarrow$ | KID $\downarrow$ | Recall $\uparrow$ | IS $\uparrow$ | FID $\downarrow$ | KID $\downarrow$ | Rec $\uparrow$ | IS $\uparrow$ |
| DDPM (Ho et al., 2020) | 10.163 | 0.0029 | 0.46 | **13.45** | 9.363 | 0.0032 | 0.47 | **14.27** |
| +ADA (Karras et al., 2020) | 9.482 | 0.0032 | 0.51 | 12.44 | 8.927 | 0.0033 | 0.51 | 12.89 |
| +RS (Mahajan et al., 2018) | 11.432 | 0.0038 | 0.44 | 12.12 | 10.259 | 0.0037 | 0.47 | 12.38 |
| +Focal (Lin et al., 2017b) | 10.212 | 0.0032 | 0.47 | 13.07 | 9.477 | 0.0034 | 0.49 | 13.31 |
| CBDM (Qin et al., 2023) | 10.051 | 0.0036 | 0.51 | 12.35 | 8.946 | 0.0036 | **0.55** | 12.59 |
| OC (Zhang et al., 2024) | 8.309 | 0.0026 | **0.52** | 13.44 | 7.188 | 0.0024 | 0.54 | 13.99 |
| CM | **7.519** | **0.0017** | **0.52** | **13.45** | **6.732** | **0.0021** | **0.55** | 14.12 |

Table 3: Per-split FIDs ($\downarrow$) of CM and baselines on Imb. CIFAR-10 (IR $= 100$) and Imb. CIFAR-100 (IR $= 100$) shown. Many, Medium, and Few are the three splits based on training imbalancedness. **Best** and second-best results are highlighted. *Results with Mean$\pm$Std can be found in Tab. G.4.*

| | Imb. CIFAR-10 (IR $= 100$) | | | | Imb. CIFAR-100 (IR $= 100$) | | | |
|---|---|---|---|---|---|---|---|---|
| Method | Many FID $\downarrow$ | Med. FID $\downarrow$ | Few FID $\downarrow$ | Overall FID $\downarrow$ | Many FID $\downarrow$ | Med. FID $\downarrow$ | Few FID $\downarrow$ | Overall FID $\downarrow$ |
| DDPM (Ho et al., 2020) | 14.203 | 19.714 | 15.869 | 10.697 | 14.068 | 15.660 | 22.188 | 10.163 |
| CBDM (Qin et al., 2023) | 12.222 | 14.465 | 12.230 | 8.2334 | 12.585 | 14.042 | 20.667 | 10.051 |
| OC (Zhang et al., 2024) | 12.026 | 15.234 | 12.254 | 8.3896 | 11.731 | 13.133 | 21.053 | 8.309 |
| CM | **11.705** | **14.340** | **11.218** | **7.7273** | **11.713** | **13.043** | **18.729** | **7.519** |

extracted from a pre-trained Inception-V3 (Szegedy et al., 2016) model[3]. During evaluation, the metrics are calculated using a balanced set of real images and 50,000 generated images. The metrics for each {*many*, *medium*, *few*} split are computed using the corresponding split's real images and 20,000 generated images. To mitigate discrepancies from the **high sensitivity** of metrics like FID to implementation details (*e.g.*, sample count, image preprocessing, input normalization, and feature dimensionality), we benchmark all methods under a unified evaluation pipeline. We stress that our models are trained on imbalanced datasets, which are **more challenging** and results in higher baseline FIDs. Accordingly, our analysis prioritizes the relative performance gains within this challenging regime over absolute FID values reported on balanced benchmarks.

**Implementation details.** We provide comprehensive implementation details in Sec. G.1.

## 4.2 MAIN RESULTS

**Performance on Imb. CIFAR-10 and CIFAR-100.** In Tab. 2, we summarize the FIDs, KIDs, Recalls, ISs of CM and all baselines on Imb. CIFAR-10 and Imb. CIFAR-100 with different imbalance ratios IR $= \{50, 100\}$. CM achieves the best results on 16 metrics across all four settings, except for two slightly lower ISs. Note that IS cannot detect mode collapse (Barratt and Sharma, 2018), *e.g.,* if the generated minority samples are overwhelmed by majority characteristics, such low-quality images would not lead to a drop in IS, which explains why vanilla DDPM still performs well on some ISs. Additionally, IS lacks a reference to real images, making it generally considered a less reliable metric (Borji, 2019; Nunn et al., 2021). On the most widely used metric FID, CM achieve significant improvements over DDPM with gains of 2.725, 2.844, 2.644, and 2.571, and consistent improvements over the best baseline in each setting by 0.506, 0.561, 0.790, and 0.456. Additionally, we provide a direct comparison with the concurrent work Overlap Optimization (Yan et al., 2024) in Sec. G.10, where CM consistently achieves superior performance.

---

[3] https://github.com/toshas/torch-fidelity/releases/download/v0.2.0/weights-inception-2015-12-05-6726825d.pth

Table 4: FIDs ($\downarrow$), KIDs ($\downarrow$), and per-class FIDs ($\downarrow$) of CM and baselines on Imb. CelebA-HQ with imbalance ratios IR $= \{100, 50\}$ shown side-by-side. Female and Male are the two classes. **Best** and second-best results are highlighted. *Results with Mean±Std can be found in Tab. G.2.*

| Imb. CelebA-HQ | IR $= 100$ | | | | IR $= 50$ | | | |
|---|---|---|---|---|---|---|---|---|
| Method | Female FID $\downarrow$ | Male FID $\downarrow$ | Overall FID $\downarrow$ | KID $\downarrow$ | Female FID $\downarrow$ | Male FID $\downarrow$ | Overall FID $\downarrow$ | KID $\downarrow$ |
| DDPM (Ho et al., 2020) | 7.143 | 16.425 | 8.727 | 0.0037 | 7.348 | 14.808 | 8.007 | 0.0034 |
| CBDM (Qin et al., 2023) | 7.043 | 14.273 | 7.823 | 0.0043 | 7.317 | 12.592 | 7.423 | 0.0042 |
| OC (Zhang et al., 2024) | 7.092 | 13.962 | 7.871 | 0.0034 | 7.283 | 12.938 | 7.438 | 0.0034 |
| CM | **6.815** | **12.788** | **7.538** | **0.0033** | **7.147** | **11.273** | **7.193** | **0.0033** |

**Many/Medium/Few analysis.** In Tab. 3, we show the fine-grained {*many*, *medium*, *few*} per-split FIDs of different methods on Imb. CIFAR-10 and Imb. CIFAR-100 with imbalance ratio IR $=$ 100. Our method achieves the best results across all three splits, with the primary improvements observed in the Medium and Few classes. It is noteworthy that on Imb. CIFAR-10, the generation quality for Medium classes is worse than for Few classes. Similar observations have been made on imbalanced contrastive learning (Zhou et al., 2023a). This could be attributed to the inherent difficulty differences between classes, suggesting a promising direction of addressing imbalanced diffusion models by combining inherent difficulty imbalance with quantity imbalance.

**Performance on Imb. CelebA-HQ.** In Tab. 4, we report the FIDs, KIDs, and per-class FIDs of CM and baselines on Imb. CelebA-HQ with different imbalance ratios IR $= \{100, 50\}$. Imb. CelebA-HQ contains two classes: Female and Male, with Female being the majority class. CM achieves the best performance across all eight metrics in both settings. Specifically, it improves the Overall FID by 1.189 and 0.814 compared to DDPM and by 0.285 and 0.230 compared to the best baselines in each setting. For the minority class (Male), CM enhances FID by 3.637 and 3.535 over DDPM and by 1.174 and 1.319 over the best baselines. In Fig. G.6, we showcase the generated results for the "Male" class (minority) with IR $=$ 100. CM generates more realistic and diverse faces.

**Performance on datasets with thousands of classes (ImageNet-LT and iNaturalist).** In Tab. 5, we report results on ImageNet-LT and iNaturalist, large-scale datasets featuring thousands of classes (1,000 and 8,142, respectively) and large-scale datasets featuring thousands of classes and extreme imbalance, specifically **ImageNet-LT** (IR $=$ 256) and **iNaturalist** (IR $=$ 500). This verifies CM's robustness to imbalance ratios significantly beyond 100:1. Following Bartosh et al. (2024); Huang et al. (2024), we adopt resolutions of 32×32 and 64×64. While the baseline CBDM shows performance degradation compared to DDPM when handling thousands of classes, exhibiting limitations at this scale, our proposed method still demonstrates significant and consistent advantages over all baselines.

**Performance of Fine-tuning Stable Diffusion on Imb. ArtBench-10.** On Imb. ArtBench-10, we fine-tune the Stable Diffusion model[4] (Rombach et al., 2022) by LoRA (Hu et al., 2022) with a rank of 128. And for $\theta^e$, the rank is set to 8. We train the model in a class-conditional manner where the textual prompt is simply set as "a {class} painting" such as "a renaissance painting". The dropout rate is set to 0.1, and the model is trained for 100 epochs with a batch size of 64, using the AdamW optimizer (Loshchilov and Hutter, 2019) with a weight decay of $10^{-6}$ and an initial learning rate of $3 \times 10^{-4}$. During inference, we generate new images using a 50-step DDIM solver (Song et al., 2021a). In Tab. 6, we compare our CM against DDPM and the two strongest baselines, CBDM and OC, on Imb. ArtBench-10 with imbalance ratios IR $= \{100, 50\}$. Our CM achieves the best results across all eight metrics. The generated images for one of the few classes "Realism" on Imb. ArtBench-10 with IR $=$ 100 are shown in Fig. 3. Our method generates more diverse images, and the generated styles are closer to the real images. This visual diversity is quantitatively supported by the superior Recall scores (Tab. 6), and we provide extensive visualizations of the intra-class diversity for tail classes in Fig. G.5. See Sec. G.8 for more visualization results.

## 4.3 FURTHER ANALYSIS

**Human and Downstream Evaluations.** In addition to the above automatic metrics, we also present human preference studies (Sec. G.4) and downstream utility assessments (Sec. G.5).

---

[4]https://huggingface.co/lambdalabs/miniSD-diffusers

Table 5: FIDs ($\downarrow$) and KIDs ($\downarrow$) on ImageNet-LT and iNaturalist at $32\times32$ and $64\times64$ resolutions.

| Method | ImageNet-LT $32\times32$ | | ImageNet-LT $64\times64$ | | iNaturalist $32\times32$ | | iNaturalist $64\times64$ | |
|---|---|---|---|---|---|---|---|---|
| | FID $\downarrow$ | KID $\downarrow$ | FID $\downarrow$ | KID $\downarrow$ | FID $\downarrow$ | KID $\downarrow$ | FID $\downarrow$ | KID $\downarrow$ |
| DDPM (Ho et al., 2020) | 13.42 | 0.0082 | 8.84 | 0.0028 | 15.66 | 0.0095 | 7.99 | 0.0030 |
| CBDM (Qin et al., 2023) | 14.66 | 0.0096 | 11.71 | 0.0068 | 18.01 | 0.0136 | 10.28 | 0.0044 |
| OC (Zhang et al., 2024) | 12.61 | 0.0075 | 8.43 | 0.0025 | 14.29 | 0.0092 | 7.67 | 0.0023 |
| CM | **10.94** | **0.0056** | **7.72** | **0.0023** | **13.38** | **0.0089** | **6.82** | **0.0020** |

Table 6: FIDs ($\downarrow$), KIDs ($\downarrow$), Recalls ($\uparrow$), and ISs ($\uparrow$) of CM and various baselines on Imb. ArtBench-10 with IR $= \{100, 50\}$ shown, using LoRA to fine-tune Stable Diffusion. **Best** and second-best results are highlighted. *Full results with Mean$\pm$Std can be found in Tab. G.3.*

| Imb. ArtBench-10 | IR $= 100$ | | | | IR $= 50$ | | | |
|---|---|---|---|---|---|---|---|---|
| Method | FID $\downarrow$ | KID $\downarrow$ | Recall $\uparrow$ | IS $\uparrow$ | FID $\downarrow$ | KID $\downarrow$ | Recall $\uparrow$ | IS $\uparrow$ |
| DDPM (Ho et al., 2020) | 27.083 | 0.0142 | 0.39 | 8.47 | 25.557 | 0.0134 | 0.39 | 8.41 |
| CBDM (Qin et al., 2023) | 25.723 | 0.0122 | 0.43 | 7.97 | 24.487 | 0.0114 | 0.43 | 8.03 |
| OC (Zhang et al., 2024) | 24.315 | 0.0106 | 0.42 | **8.71** | 23.287 | 0.0097 | 0.43 | 8.48 |
| CM | **22.776** | **0.0087** | **0.44** | **8.71** | **21.733** | **0.0080** | **0.44** | **8.51** |

Imb. ArtBench-10, IR = 100, Realism

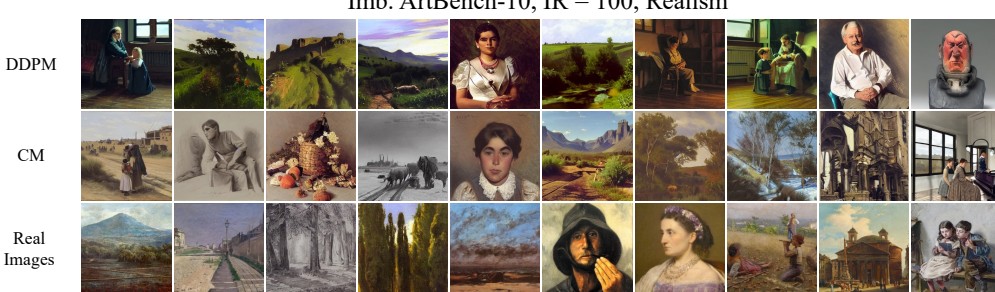

Figure 3: The visualization of generated images for the class "Realism", which is one of the few classes on Imb. ArtBench-10 with IR $= 100$. The last row displays real images from the dataset for reference. It is evident that CM generates results that are significantly more diverse and stylistically closer to the real images compared to DDPM. The images shown are randomly selected.

**CM as a universal framework.** Fig. 4(a) summarizes the performance of our CM when integrated with various baselines (*i.e.,* using the corresponding objective function for $\mathcal{L}_{\text{base}}$ in Eq. (4)) on Imb. CIFAR-100 (IR $= 100$). CM consistently improves the performance of imbalanced generation when combined with different baselines. Due to the orthogonality of CM to existing methods, it can consistently benefit from improved objective functions, including potential future advancements.

**Knowledge allocation between $\theta^g$ and $\theta^e$.** We evaluate knowledge allocation by comparing our full model CM (using $\theta = \theta^g \oplus \theta^e$) with an ablation using only $\theta^g$ (denoted CM ($\theta^g$)). On Imb. CIFAR-100 (IR $= 100$; Tab. 7), CM ($\theta^g$) excels on Many and Medium classes but falters on Few classes. In contrast, CM shows strong performance across all splits. This indicates that CM successfully allocates minority expertise to $\theta^e$, while reserving majority and general knowledge for $\theta^g$.

**Ablation on the hyperparameter $\lambda$ in Eq. (4).** We conduct ablation experiments on the hyperparameter $\lambda$, the weight of the CM loss in Eq. (4). Fig. 4(b) illustrates how the FID of CM changes with varying $\lambda$ values. We observe that CM maintains a consistent advantage over OC across a wide range of $\lambda$ values, with its performance peaking around $\lambda = 1.0$.

**Ablation on the rank of $W^e \in \theta^e$.** We conduct an ablation study on the rank ratio of $W^e \in \theta^e$ relative to full rank, *i.e.,* $\frac{r}{\min(d,k)}$. Fig. 4(c) shows how the FID varies with different rank ratios. CM consistently outperforms OC across a wide range of rank ratios, with peak performance around 0.1.

**Ablation on $\mathcal{L}_{\text{Con}}$ and $\mathcal{L}_{\text{Div}}$.** Tab. 7 presents the results of CM as well as the ablation study where the consistency loss $\mathcal{L}_{\text{Con}}$ and the diversity loss $\mathcal{L}_{\text{Div}}$ are removed separately from CM. Since $\mathcal{L}_{\text{Con}}$

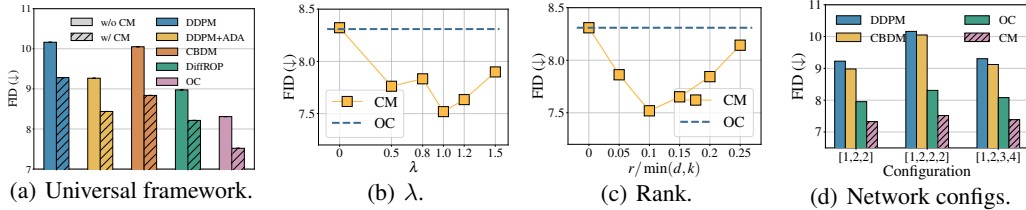

(a) Universal framework.  (b) $\lambda$.  (c) Rank.  (d) Network configs.

Figure 4: (a) The performance of CM when integrated with baselines. (b) Ablation study on the hyperparameter $\lambda$ in Eq. (4). (c) Ablation study on the rank $r$. (d) Ablation study on various UNet configurations. All experiments are conducted on Imb. CIFAR-100 with IR $= 100$. In (b) and (c), we use OC as a reference because it shows the best overall performance among the baselines.

Table 7: (Left) Per-split FIDs and overall FIDs ($\downarrow$) of DDPM, CM ($\theta^g$), and CM on Imb. CIFAR-100 with imbalance ratio IR $= 100$. (Right) FIDs ($\downarrow$), KIDs ($\downarrow$), Recalls ($\uparrow$), and ISs ($\uparrow$) on Imb. CIFAR-100 with imbalance ratio IR $= 100$. The last two rows show the results of CM after removing $\mathcal{L}_{\text{Con}}$ and $\mathcal{L}_{\text{Div}}$, respectively. *Full results with Mean$\pm$Std can be found in Tables G.5 and G.6.*

| Method | Many FID $\downarrow$ | Medium FID $\downarrow$ | Few FID $\downarrow$ | Overall FID $\downarrow$ |
|---|---|---|---|---|
| DDPM (Ho et al., 2020) | 14.068 | 15.660 | 22.188 | 10.163 |
| CM ($\theta^g$) | 11.923 | 14.872 | 29.357 | 13.712 |
| CM ($\theta = \theta^g \oplus \theta^e$) | **11.713** | **13.043** | **18.729** | **7.519** |

| Method | FID $\downarrow$ | KID $\downarrow$ | Recall $\uparrow$ | IS $\uparrow$ |
|---|---|---|---|---|
| DDPM (Ho et al., 2020) | 10.163 | 0.0029 | 0.46 | **13.45** |
| OC (Zhang et al., 2024) | 8.309 | 0.0026 | **0.52** | 13.44 |
| CM | 7.519 | **0.0017** | **0.52** | 13.45 |
| CM w/o $\mathcal{L}_{\text{Con}}$ | 8.412 | 0.0029 | 0.50 | 13.23 |
| CM w/o $\mathcal{L}_{\text{Div}}$ | 8.073 | 0.0025 | 0.51 | 13.42 |

and $\mathcal{L}_{\text{Div}}$ are responsible for allocating majority knowledge and minority expertise, respectively, removing either leads to a significant drop in performance, highlighting their necessity.

**Ablation on network configurations.** We conduct experiments on UNet architectures with varying widths and depths, which is achieved by setting the channel_multipliers parameter to $[1, 2, 2]$, $[1, 2, 2, 2]$ (default), and $[1, 2, 3, 4]$. Fig. 4(d) shows FIDs on Imb. CIFAR-10 and CIFAR-100 with IR $= 100$. CM consistently demonstrates clear advantages across different network configurations.

**Broader settings.** The main experiments in this paper are conducted on the classical DDPM network (trained from scratch) and Stable Diffusion (fine-tuned), where controlled comparisons demonstrate and analyze the effectiveness of CM on class-imbalanced data. In addition, we validate our method in broader settings, including consistency models, integration with DPM-Solver, training a latent diffusion model from scratch, and fine-grained text-to-image datasets, detailed in Sec. G.9. Furthermore, we explore the potential of CM in discriminative tasks. On Long-Tailed CIFAR-100 classification using ResNet-18, applying CM to strong baselines (LDAM (Cao et al., 2019a) and Logit Adjustment (Menon et al., 2021a)) yields consistent improvements (*e.g.*, +1.78% accuracy for LA at IR $= 100$), suggesting that our capacity reservation principle generalizes beyond generative modeling. Detailed results are provided in Sec. G.11.

## 5 CONCLUSION

In this study, we seek to improve the robustness of diffusion models to class-imbalanced data. Unlike previous work that focuses on improving objective functions, we aim to protect the generation performance of minorities by reserving and allocating model capacity for them. We first decompose the model parameters into parts that capture general and majority knowledge, and a dedicated part for minority expertise using low-rank decomposition techniques. By introducing a capacity manipulation loss, we successfully allocate the corresponding knowledge to the reserved model capacity during training. Extensive experiments and empirical analyses confirm that our CM effectively protects minorities in imbalanced diffusion models via capacity manipulation.

ACKNOWLEDGEMENTS

This work is supported by National Key R&D Program of China (No. 2022ZD0160702), National Natural Science Foundation of China (No. 62306178), STCSM (No. 22DZ2229005), and 111 plan (No. BP0719010).

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

# APPENDIX: IMPROVING DIFFUSION MODELS FOR CLASS-IMBALANCED TRAINING DATA VIA CAPACITY MANIPULATION

## CONTENTS

## A  BACKGROUNDS ON DIFFUSION MODELS

We briefly review discrete-time diffusion models, specifically denoising diffusion probabilistic models (DDPMs) (Ho et al., 2020). Given a random variable $\mathbf{x} \in \mathcal{X}$ and a *forward diffusion process* on $\mathbf{x}$ defined as $\mathbf{x}_{1:T} := \mathbf{x}_1, \ldots, \mathbf{x}_T$ with $T \in \mathbb{N}^+$, the Markov transition probability from $\mathbf{x}_{t-1}$ to $\mathbf{x}_t$ is $q(\mathbf{x}_t|\mathbf{x}_{t-1}) = \mathcal{N}(\mathbf{x}_t; \sqrt{1 - \beta_t}\mathbf{x}_{t-1}, \beta_t\mathbf{I})$, where $\mathbf{x}_0 := \mathbf{x} \sim q(\mathbf{x}_0)$, and $\{\beta_t\}_{t=1}^T$ is the variance schedule. The forward process allows us to sample $\mathbf{x}_t$ at an arbitrary timestep $t$ directly from $\mathbf{x}_0$ in a closed form $q(\mathbf{x}_t|\mathbf{x}_0) = \mathcal{N}(\mathbf{x}_t; \sqrt{\bar{\alpha}_t}\mathbf{x}_0, (1 - \bar{\alpha}_t)\mathbf{I})$, where $\alpha_t := 1 - \beta_t$ and $\bar{\alpha}_t := \prod_{i=1}^t \alpha_i$. The variance schedule is prescribed such that $\mathbf{x}_T$ is nearly an isotropic Gaussian distribution.

**Training objective.** The *reverse process* is defined as a Markov chain that aims to approximate $q(\mathbf{x}_0)$ by gradually denoising from the standard Gaussian distribution $p(\mathbf{x}_T) = \mathcal{N}(\mathbf{x}_T; \mathbf{0}, \mathbf{I})$: $p_\theta(\mathbf{x}_{t-1}|\mathbf{x}_t) = \mathcal{N}(p_\theta(\mathbf{x}_{t-1}; \boldsymbol{\mu}_\theta(\mathbf{x}_t, t), \sigma_t^2\mathbf{I})$, where $\boldsymbol{\mu}_\theta(\mathbf{x}_t, t) = \frac{1}{\sqrt{\alpha_t}}(\mathbf{x}_t - \frac{\beta_t}{\sqrt{1-\bar{\alpha}_t}}\boldsymbol{\epsilon}_\theta(\mathbf{x}_t, t))$ is parameterized by a time-conditioned noise prediction network $\boldsymbol{\epsilon}_\theta(\mathbf{x}_t, t)$ and $\sigma_1, \ldots, \sigma_T$ are time dependent constants that can be predefined or analytically computed (Bao et al., 2022). The reverse process can be learned by optimizing the variational lower bound on log-likelihood as

$$\log p_\theta(\mathbf{x}) \geq \mathbb{E}_q[-D_{\mathrm{KL}}(q(\mathbf{x}_T|\mathbf{x}_0)\|p(\mathbf{x}_T)) + \log p_\theta(\mathbf{x}_0|\mathbf{x}_1) - \sum_{t>1} D_{\mathrm{KL}}(q(\mathbf{x}_{t-1}|\mathbf{x}_t, \mathbf{x}_0)\|p_\theta(\mathbf{x}_{t-1}|\mathbf{x}_t))]$$

$$= -\mathbb{E}_{\boldsymbol{\epsilon}, t}[w_t\|\boldsymbol{\epsilon}_\theta(\mathbf{x}_t, t) - \boldsymbol{\epsilon}\|_2^2] + C_1,$$

$$(A.1)$$

where $\boldsymbol{\epsilon} \sim \mathcal{N}(\boldsymbol{\epsilon}; \mathbf{0}, \mathbf{1})$, $\mathbf{x}_t = \sqrt{\bar{\alpha}_t}\mathbf{x}_0 + \sqrt{1 - \bar{\alpha}_t}\boldsymbol{\epsilon}$ according to the forward process, $w_t = \frac{\beta_t^2}{2\sigma_t^2\alpha_t(1-\bar{\alpha}_t)}$, and $C_1$ is typically small and can be dropped (Ho et al., 2020; Song et al., 2021b). The term $\mathcal{L}_{\mathrm{Diff}}(\mathbf{x}, \theta) = \mathbb{E}_{\boldsymbol{\epsilon}, t}[w_t\|\boldsymbol{\epsilon}_\theta(\mathbf{x}_t, t) - \boldsymbol{\epsilon}\|_2^2]$ is called the *diffusion loss* (Kingma et al., 2021). To benefit sample quality, Ho et al. (2020) apply a simplified training objective by setting $w_t = 1$.

**Class-conditional diffusion models.** When the class labels of the training set are available, the class-conditional diffusion model $p_\theta(\mathbf{x}|y)$ can be parameterized by $\boldsymbol{\epsilon}(\mathbf{x}_t, t, y)$. And the unconditional diffusion model $p_\theta(\mathbf{x})$ can be viewed as a special case with a null condition $\boldsymbol{\epsilon}(\mathbf{x}_t, t, \mathrm{Null})$. A similar lower bound on the class-conditional log-likelihood to Eq. (A.1) is

$$\log p_\theta(\mathbf{x}|y) \geq -\mathbb{E}_{\boldsymbol{\epsilon}, t}[w_t\|\boldsymbol{\epsilon}_\theta(\mathbf{x}_t, t, y) - \boldsymbol{\epsilon}\|_2^2] + C_2, \qquad (A.2)$$

where $C_2$ is another small constant and can be dropped (Ho et al., 2020; Song et al., 2021b). The class-conditional diffusion loss can be written as $\mathcal{L}_{\mathrm{Diff}}(\mathbf{x}, y, \theta) = \mathbb{E}_{\boldsymbol{\epsilon}, t}[w_t\|\boldsymbol{\epsilon}_\theta(\mathbf{x}_t, t, y) - \boldsymbol{\epsilon}\|_2^2]$.

## B  RELATED WORK

**Diffusion Models.** Inspired by non-equilibrium thermodynamics (Sohl-Dickstein et al., 2015), diffusion models have achieved remarkably effective performance at image generation (Ho et al., 2020; Dhariwal and Nichol, 2021; Rombach et al., 2022). Ho et al. (2020) conduct the training of diffusion models using a weighted variational bound. (Song et al., 2021b) propose an alternative method for constructing diffusion models by using a stochastic differential equation (SDE). Karras et al. (2022) introduce a design space that clearly outlines the key design choices in previous works. Denoising diffusion implicit models (DDIMs) (Song et al., 2021a) employs an alternative non-Markovian generation process, enabling faster sampling for diffusion models.

**Imbalanced Generation.** Addressing class imbalance in generative modeling was initially explored with Generative Adversarial Networks (GANs) (Goodfellow et al., 2014). For instance, CB-GAN (Rangwani et al., 2021) employed a pre-trained classifier to mitigate imbalance. Rangwani et al. (2022) attributed performance decline in long-tailed GAN generation to minority-class mode collapse—caused by spectral explosion of conditioning parameters—and proposed a group spectral regularizer to address it. More recently, with the demonstrated success of diffusion models (DMs), research has focused on their robustness to imbalanced data. CBDM (Qin et al., 2023) applies a distribution adjustment regularizer for minority class augmentation. Yan et al. (2024) introduced contrastive regularization to bolster minority representations, and OC (Zhang et al., 2024) utilizes transfer learning from majority to minority classes to improve minority generation quality.

**Imbalanced Classification.** Imbalanced Classification has been extensively studied to address long-tailed distributions in discriminative tasks (Zhang et al., 2023; 2025; Shi et al., 2024; Wei et al., 2024;

Wei and Gan, 2023). Existing literature generally tackles this challenge through three paradigms. At the data level, strategies aim to mitigate imbalance through re-sampling or feature synthesis, ranging from classic techniques like SMOTE (Chawla et al., 2002) to advanced data augmentations such as Mixup (Zhang et al., 2017) and Context-Rich Minority Oversampling (Park et al., 2022). At the model level, research focuses on architectural innovations (Yang et al., 2026; Luo et al., 2025) and feature enhancement (Li et al., 2024; Hong et al., 2024; Bao et al., 2025); a prominent milestone is the two-stage decoupling of representation learning and classifier training (Kang et al., 2020), which inspired dual-branch architectures like BBN (Zhou et al., 2020) to balance generalized feature extraction with unbiased classification. Finally, at the objective level, methods directly penalize majority class dominance or calibrate output confidence. This encompasses cost-sensitive re-weighting approaches like Focal Loss (Lin et al., 2017a) and Class-Balanced Loss (Cui et al., 2019a), as well as margin-based calibration (Cao et al., 2019b; Hong et al., 2026; Zhou et al., 2023b; Wang et al., 2025) and rigorous Logit Adjustment frameworks (Menon et al., 2021b; Li et al., 2022; Hong et al., 2023). While these methods are inherently designed for discriminative tasks, their underlying principles of distribution smoothing and bias calibration deeply inspire our approach to imbalanced generation.

**Reweighting and Data Mixture Optimization.** Beyond loss design for specific imbalanced classes, recent research has explored reweighting strategies and data mixture optimization to enhance generative models (Xie et al., 2023; Fan et al., 2024; Kim et al., 2024; Li et al., 2025; Liu et al., 2025). In the context of large language models, methods like DoReMi (Xie et al., 2023) and RegMix (Liu et al., 2025) optimize the weights of data domains to improve pretraining efficiency and performance. DoGE (Fan et al., 2024) further introduces domain reweighting based on generalization estimation. For diffusion models, Kim et al. (2024) propose importance weighting to train unbiased models from biased datasets, while Li et al. (2025) combine structural pruning with data reweighting for efficient training. While these approaches focus on adjusting the importance or sampling frequency of training data, our proposed Capacity Manipulation (CM) takes an orthogonal approach by explicitly reserving and allocating learnable parameters (model capacity) for minority concepts, ensuring they are not overwhelmed by majority updates.

## C  ALGORITHM PSEUDOCODE

We summarize the procedure of our CM in Algorithm C.1, where we use DDPM as the base loss, employ DDPM for sampling, and illustrate the process in a sample-wise manner as an example.

---

**Algorithm C.1** Algorithm of CM

---

$\triangleright$ Training, take DDPM as base, sample-wise
**Initialize**: $\theta^g = \{W_1^g, W_2^g, \ldots\}$, $\theta^e = \{B_1^e A_1^e, B_2^e A_2^e, \ldots\}$
**repeat**
 Sample data $(\mathbf{x}, y) \in \mathcal{D}$
 Sample a timestep $t \sim \text{Uniform}(\{1, \ldots, T\})$
 Sample a noise $\boldsymbol{\epsilon} \sim \mathcal{N}(\mathbf{0}, \mathbf{I})$
 Base loss: $\mathcal{L}_{\text{base}} = \|\boldsymbol{\epsilon}_{\theta^g \oplus \theta^e}(\sqrt{\bar{\alpha}_t}\mathbf{x} + (1 - \bar{\alpha}_t)\boldsymbol{\epsilon}, t, y) - \boldsymbol{\epsilon}\|_2^2$
 Capacity allocation loss: $\mathcal{L}_{\text{CM}} = (\omega_{\text{Con}}^y - \omega_{\text{Div}}^y)\|\boldsymbol{\epsilon}_{\theta^g \oplus \theta^e}(\mathbf{x}_t, t, y) - \boldsymbol{\epsilon}_{\theta^g}(\mathbf{x}_t, t, y)\|_2^2$.
 Take gradient descent on $\nabla_{\theta^g, \theta^e}(\mathcal{L}_{\text{base}} + \lambda \mathcal{L}_{\text{CM}})$
**until** converged
$\triangleright$ Sampling, take DDPM for example, sample-wise
Merge model parameters as $\theta = \theta^g \oplus \theta^e$
Sample $\mathbf{x}_T \sim \mathcal{N}(\mathbf{0}, \mathbf{I})$
**for** $t = T, \ldots, 1$ **do**
 $\mathbf{z} \sim \mathcal{N}(\mathbf{0}, \mathbf{I})$ if $t > 1$, else $z = \mathbf{0}$
 $\mathbf{x}_{t-1} = \frac{1}{\sqrt{\alpha_t}}(\mathbf{x}_t - \frac{\beta_t}{\sqrt{1 - \bar{\alpha}_t}}\boldsymbol{\epsilon}_\theta(\mathbf{x}_t, t, y)) + \sigma_t \mathbf{z}$
**end for**
**return** $\mathbf{x}_0$

---

## D  BRIEF INTRODUCTION TO THE OBJECTIVES OF CBDM AND OC

Here, we briefly introduce the objective functions of CBDM (Qin et al., 2023) and OC (Zhang et al., 2024).

**CBDM.**  The objective loss function of CBDM is defined as:

$$\mathcal{L}_{\text{CBDM}}(\mathcal{D}, \theta) = \frac{1}{N} \sum_{\text{x}, y \in \mathcal{D}} \mathbb{E}_{\epsilon, t}[\|\epsilon_{\theta}(\text{x}_t, t, y) - \epsilon\|_2^2 + \frac{\tau t}{|\mathcal{Y}|} \sum_{y' \in \mathcal{Y}} (\|\epsilon_{\theta}(\text{x}_t, t, y)$$
$$- \text{sg}(\epsilon_{\theta}(\text{x}_t, t, y'))\|_2^2 + \gamma \|\text{sg}(\epsilon_{\theta}(\text{x}_t, t, y)) - \epsilon_{\theta}(\text{x}_t, t, y')\|_2^2)]$$
(D.1)

where $\text{sg}(\cdot)$ denotes the stop gradient operation; $\tau$ and $\gamma$ are hyperparameters. $|\mathcal{Y}|$ is the class number. CBDM introduces a regularizer to the diffusion loss to balance the generation quality across different classes.

**OC.**  The objective loss function of CBDM is defined as:

$$\mathcal{L}_{\text{OC}}(\mathcal{D}, \theta) = \frac{1}{N} \sum_{\text{x}, y \in \mathcal{D}} \mathbb{E}_{\epsilon, t} \left[ \|\epsilon_{\theta}(\text{x}_t, t, y) - \hat{\epsilon}\|_2^2 \right],$$
(D.2)

where $\hat{\epsilon}$ represents a knowledge transfer term within a batch of samples. For any given $(\text{x}, y) \in \mathcal{D}$ and its corresponding randomly sampled $\epsilon$ and $t$ during training, OC selects a reference sample $(\hat{\text{x}}, \hat{y})$ from the same batch. The term $\hat{\epsilon}$ is then defined as:

$$\hat{\epsilon} = \begin{cases} \epsilon, & \text{if } q(\hat{y}) < q(y), \\ \frac{\text{x}_t}{\sqrt{\bar{\alpha}_t}} - \hat{\text{x}}, & \text{if } q(\hat{y}) \geq q(y). \end{cases}$$
(D.3)

Here, the probability of any $(\hat{\text{x}}, \hat{y}) \in \text{Batch}(\text{x}, y)$ (where $\text{Batch}(\text{x}, y)$ represents the batch containing the sample $(\text{x}, y)$) being selected as the reference sample is given by:

$$p_{\text{sel}}(\hat{\text{x}}, \hat{y}) = \frac{q(\text{x}_t | \hat{\text{x}}, \hat{y})}{\sum_{\text{x}', y' \in \text{Batch}(\text{x}, y)} q(\text{x}_t | \text{x}', y')}.$$
(D.4)

By setting $\mathcal{L}_{\text{base}}$ to $\mathcal{L}_{\text{CBDM}}$ or $\mathcal{L}_{\text{OC}}$, our method CM can be seamlessly integrated with these improved objective functions.

## E  MORE DISCUSSION

### E.1  DEFINITION AND DISCUSSION OF MODEL CAPACITY

To clarify the concept of "capacity" used throughout the paper, we define it as the *available representational resources* of the neural network to capture and store data features. We operationalize this concept from two complementary perspectives:

- Structural Perspective (Rank): Following the principles of Low-Rank Adaptation (Hu et al., 2022), we treat the rank of weight matrices as a direct measure of capacity. A full-rank matrix possesses maximum capacity, whereas a low-rank decomposition restricts information flow. Our method explicitly manipulates this by reserving a specific rank $r$ in $\theta^e$ for minority expertise.
- Optimization Perspective (Gradient Dominance): Theoretically, capacity allocation is determined by which classes dominate the parameter updates. As proven in Theorem 2.1, majority classes contribute significantly larger aggregated gradients, thereby dominating the update direction $\Delta W$ and effectively monopolizing the parameters.

In Sec. 2.2, we utilize the L1-norm magnitude solely as a diagnostic proxy rather than a definition. Since parameters with larger magnitudes typically contribute more to the model's output, the sensitivity of minority classes to the pruning of small-magnitude parameters empirically indicates that they are forced into "weak" connections (marginal capacity).

### E.2 COMPARISON WITH ENSEMBLE-BASED IMBALANCED CLASSIFICATION METHODS.

Several ensemble-based methods (Cui et al., 2023; Wang et al., 2021b; Zhang et al., 2022) leverage multiple experts to capture diverse knowledge, achieving strong performance in classification tasks through prediction ensemble. However, most of these methods are tailored for classification networks in terms of architecture, training paradigm, and loss functions, making them unsuitable for direct application in diffusion models. While they also involve knowledge allocation, their gain mainly comes from increased capacity and ensemble predictions. Additionally, they often require structural modifications to the network and incur higher inference latency, further limiting applicability. In contrast, our method introduces no changes to network structure, does not increase model capacity or inference latency, and enhances imbalanced diffusion models purely through capacity manipulation.

### E.3 CONCEPTUAL DISTINCTIONS

To clarify the position of Capacity Manipulation (CM) within the broader landscape of long-tailed learning and efficient tuning, we provide a side-by-side comparison in Tab. E.1. Unlike Mixture-of-Experts (MoE) or Class-Specific Adapters, which often incur high inference overhead due to dynamic routing or loading specific weights, CM employs a joint training paradigm with zero inference overhead (via merged weights). Furthermore, unlike reweighting or class-balanced objectives that focus solely on loss design, CM explicitly addresses the bottleneck of model capacity allocation.

Table E.1: Conceptual comparison of CM with related paradigms.

| Method | Mechanism | Training Paradigm | Inference Overhead |
|---|---|---|---|
| Reweighting / Resampling | Data/Loss Weighting | Single Stage | None |
| Class-Balanced Objectives (e.g., OC) | Loss Design | Single Stage | None |
| Adapters / LoRA | Parameter Addition | Usually Fine-tuning | None (if merged) / Low |
| Mixture-of-Experts (MoE) | Dynamic Routing | Joint Training | High (Routing + Experts) |
| **CM (Ours)** | **Capacity Reservation** | **Joint Training** | **Zero (Merged Weights)** |

### E.4 COMPARISON WITH MOE-STYLE BASELINE (GROUP-EXPERT LORA)

To empirically verify the advantage of CM over simple parameter addition or routing strategies, we implemented a **"Group-Expert LoRA"** baseline under matched compute budgets. Specifically, we assign distinct LoRA adapters to "Many", "Medium", and "Few" class groups, effectively acting as a deterministic MoE where the router is determined by the class frequency group. We trained this baseline on Imb. CIFAR-100 ($IR = 100$) using the same rank budget as CM.

As shown in Tab. E.2, CM outperforms the Group-Expert LoRA by a large margin (FID 7.52 vs. 10.06). This indicates that simply assigning experts to different groups (MoE-style) is insufficient for addressing the capacity dominance of majority classes in diffusion models. CM's superior performance is attributed to its explicit capacity reservation mechanism via the proposed capacity manipulation loss.

Table E.2: Comparison with MoE-style Baseline on Imb. CIFAR-100 ($IR = 100$).

| Method | Architecture | FID ↓ |
|---|---|---|
| DDPM | Standard U-Net | 10.16 |
| Group-Expert LoRA (MoE-style) | 3 Experts (Many/Med/Few) | 10.06 |
| **CM (Ours)** | **Capacity Reservation** | **7.52** |

## F THEORETICAL SUPPLEMENT

### F.1 PROBLEM SETUP FOR THEOREM 2.1 AND THEOREM 3.1

We consider a simplified version of training a diffusion model on imbalanced data, involving only two classes $\mathcal{Y} = \{1, 2\}$, where $N_1 \gg N_2$. Therefore, the total loss for all samples in the training set

is:

$$\mathcal{L} = \sum_{i=1}^{N_1+N_2} \mathcal{L}_{\text{Diff}}(\mathbf{x}_i, y_i, \theta). \tag{F.1}$$

For any parameter matrix $W \in \theta$, its gradient is:

$$\begin{aligned} \nabla_W \mathcal{L} &= \sum_{i=1}^{N_1+N_2} \nabla_W \mathcal{L}_{\text{Diff}}(\mathbf{x}_i, y_i, \theta) \\ &= \sum_{i=1}^{N_1} g_1^{(i)} + \sum_{j=1}^{N_2} g_2^{(j)}, \end{aligned} \tag{F.2}$$

where $g_1^{(i)}$ and $g_2^{(j)}$ denote the gradients contributed by the $i$-th sample of class $y = 1$ and the $j$-th sample of class $y = 2$ to $W$, respectively. Here $W$ is omitted in the notation for brevity.

## F.2 PROOF OF THEOREM 2.1

**Assumption F.1.** For samples of the same class $y \in \mathcal{Y}$, their gradient contributions $g_y^{(i)}$ to $W$ follow a normal distribution with mean $\mu_y$ and variance $\sigma_y^2 I$, *i.e.*, $g_y^{(i)} \sim \mathcal{N}(\mu_y, \sigma_y^2 I)$. For gradients of different classes, their Gaussian distributions share the same magnitude of mean vectors but differ in direction, while having identical variance magnitudes, *i.e.*,

$$\|\mu_1\| = \|\mu_2\| = \mu, \quad \mu_1 \neq \mu_2, \quad \sigma_1^2 = \sigma_2^2 = \sigma^2.$$

where $\mu$ and $\sigma$ are positive constants.

*Proof of Theorem 2.1.*
According to Assumption F.1, $\nabla_W \mathcal{L}$ follows a normal distribution with:

$$\begin{aligned} \mathbb{E}[\nabla_W \mathcal{L}] &= N_1 \mu_1 + N_2 \mu_2, \\ \text{Var}[\nabla_W \mathcal{L}] &= N_1 \sigma^2 I + N_2 \sigma^2 I = (N_1 + N_2)\sigma^2 I \end{aligned} \tag{F.3}$$

The update direction is driven by the majority class $y = 1$ if the angle between the total gradient $\nabla_W \mathcal{L}$ and the expected gradient of class 1 ($\mu_1$) is smaller than the angle between G and the mean gradient of class 2 ($\mu_2$). This condition is equivalent to requiring that $\nabla_W \mathcal{L} \cdot \mu_1 > \nabla_W \mathcal{L} \cdot \mu_2$, which can be further rewritten as $\nabla_W \mathcal{L} \cdot (\mu_1 - \mu_2) > 0$.

Let $Z = \nabla_W \mathcal{L} \cdot (\mu_1 - \mu_2)$. Since $\nabla_W \mathcal{L}$ follows a multivariate normal distribution, $Z$ is a univariate normal random variable. We next compute the mean and variance of $Z$.

$$\begin{aligned} \mathbb{E}[Z] &= (N_1 \mu_1 + N_2 \mu_2) \cdot (\mu_1 - \mu_2) \\ &= N_1 \|\mu_1\|^2 - N_1 \mu_1 \cdot \mu_2 + N_2 \mu_2 \cdot \mu_1 - N_2 \|\mu_2\|^2 \\ &= N_1 \mu^2 - N_1 \mu^2 \cos\angle(\mu_1, \mu_2) + N_2 \mu^2 \cos\angle(\mu_1, \mu_2) - N_2 \mu^2 \\ &= (N_1 - N_2)\mu^2 (1 - \cos\angle(\mu_1, \mu_2)), \\ \text{Var}(Z) &= (\mu_1 - \mu_2)^\top \text{Var}(\nabla_W \mathcal{L})(\mu_1 - \mu_2) \\ &= (\mu_1 - \mu_2)^\top \left((N_1 + N_2)\sigma^2 I\right)(\mu_1 - \mu_2) \\ &= (N_1 + N_2)\sigma^2 \|\mu_1 - \mu_2\|^2 \\ &= 2(N_1 + N_2)\sigma^2 \mu^2 (1 - \cos\angle(\mu_1, \mu_2)). \end{aligned} \tag{F.4}$$

Since $Z \sim \mathcal{N}(\mathbb{E}[Z], \text{Var}(Z))$, the probability that the update direction is driven by the majority class (*i.e.*, $P(Z > 0)$) is:

$$\Pi_{\text{maj}} = P(Z > 0) = \Phi\left(\frac{\mathbb{E}[Z]}{\sqrt{\text{Var}(Z)}}\right), \tag{F.5}$$

where $\Phi(\cdot)$ is the standard normal CDF, defiend as:

$$\Phi(x) = \int_{-\infty}^{x} \frac{1}{\sqrt{2\pi}} e^{-\frac{t^2}{2}} \, dt. \tag{F.6}$$

Substituting the expressions for $\mathbb{E}[Z]$ and $\mathrm{Var}(Z)$, we have:

$$
\begin{aligned}
\Pi_{\text{maj}} &= \Phi\left(\frac{\mathbb{E}[Z]}{\sqrt{\mathrm{Var}(Z)}}\right) \\
&= \Phi\left(\frac{(N_1 - N_2)\mu^2(1 - \cos\angle(\mu_1, \mu_2))}{\sqrt{2(N_1 + N_2)\sigma^2\mu^2(1 - \cos\angle(\mu_1, \mu_2))}}\right) \\
&= \Phi\left(\frac{(N_1 - N_2)\mu\sqrt{(1 - \cos\angle(\mu_1, \mu_2))}}{\sqrt{2(N_1 + N_2)}\sigma}\right) \\
&= \Phi\left(\frac{(2a - 1)\mu\sqrt{2N(1 - \cos\angle(\mu_1, \mu_2))}}{2\sigma}\right),
\end{aligned} \tag{F.7}
$$

where $a = \frac{N_1}{N_1 + N_2}$ is the majority ratio, $N = N_1 + N_2$. $\qquad \square$

### F.3 PROOF OF THEOREM 3.1

**Assumption F.2** (Simplified Overall Update Direction for Low-Rank Modules). For the equation $W = W^g + W^e$ in Eq. (1), where $W^g$ is a full-rank learnable parameter and $W^e$ is a low-rank module with rank $r$, we model the overall update direction of $W$ using the following simplification:

$$\Delta W = \frac{\nabla_{W^g}}{\|\mathbb{E}\nabla_{W^g}\|} + \alpha_r \frac{\nabla_{W^e}}{\|\mathbb{E}\nabla_{W^e}\|}.$$

Here, the normalization terms are derived from the implicit or explicit gradient normalization commonly used in modern optimizers (Kingma and Ba, 2015; Tieleman and Hinton; Duchi et al., 2011; You et al., 2020). $\alpha_r$ is a monotonically increasing function of $r$, with $\alpha_r = 0$ when $r = 0$ and $\alpha_r = 1$ when $W^e$ is full rank. $\alpha_r$ provides a simplified model for the low-rank effect, following a general trend and strictly satisfying the boundary conditions at $r = 0$ and when $W^e$ is full rank.

*Proof of Theorem 3.1.*

$$\Pi_{\text{maj}} = P(\Delta W \cdot (\mu_1 - \mu_2) > 0) = P\left(\left(\frac{\nabla_{W^g}}{\|\mathbb{E}\nabla_{W^g}\|} + \alpha_r \frac{\nabla_{W^e}}{\|\mathbb{E}\nabla_{W^e}\|}\right) \cdot (\mu_1 - \mu_2) > 0\right) \tag{F.8}$$

Note that $\nabla_{W^g} = \sum_{i=1}^{N_1} g_1^{(i)} + \sum_{j=1}^{N_2} g_2^{(j)}$. We define a auxiliary variable $\nabla'_{W^g}$ by replacing all terms in $\nabla_W$ corresponding to class $y = 2$ with those from class $y = 1$:

$$\nabla'_{W^g} = \sum_{i=1}^{N=N_1+N_2} g_1^{(i)}. \tag{F.9}$$

This operation further amplifies the dominance probability of class $y = 1$. Therefore:

$$\Pi_{\text{maj}} < P\left(\left(\frac{\nabla'_{W^g}}{\|\mathbb{E}\nabla'_{W^g}\|} + \alpha_r \frac{\nabla_{W^e}}{\|\mathbb{E}\nabla_{W^e}\|}\right) \cdot (\mu_1 - \mu_2) > 0\right) \tag{F.10}$$

Since $\nabla'_{W^g} \sim \mathcal{N}(N\mu_1, N\sigma^2 I)$, we have:

$$
\begin{aligned}
\mathbb{E}\left[\frac{\nabla'_{W^g}}{\|\mathbb{E}\nabla'_{W^g}\|}\right] &= \frac{N\mu_1}{\|N\mu_1\|} = \frac{\mu_1}{\mu} \\
\mathrm{Var}\left(\frac{\nabla'_{W^g}}{\|\mathbb{E}\nabla'_{W^g}\|}\right) &= \frac{\sigma^2}{N\mu^2} I.
\end{aligned} \tag{F.11}
$$

Similarly, $\nabla_{W^e} \sim \mathcal{N}(N_2\mu_2, N_2\sigma^2 I)$, we have:

$$\mathbb{E}\Big[\frac{\nabla_{W^e}}{\|\mathbb{E}\nabla_{W^e}\|}\Big] = \frac{\mu_2}{\mu}$$

$$\mathrm{Var}\Big(\frac{\nabla_{W^e}}{\|\mathbb{E}\nabla_{W^e}\|}\Big) = \frac{\sigma^2}{N_2\mu^2}I. \tag{F.12}$$

Let $Z = \big(\frac{\nabla'_{W^g}}{\|\mathbb{E}\nabla'_{W^g}\|} + \alpha_r \frac{\nabla_{W^e}}{\|\mathbb{E}\nabla_{W^e}\|}\big) \cdot (\mu_1 - \mu_2)$, we have:

$$\mathbb{E}[Z] = \frac{\mu_1 + \alpha_r \mu_2}{\mu} \cdot (\mu_1 - \mu_2)$$

$$= \mu(1 - \alpha_r)(1 - \cos\angle(\mu_1, \mu_2))$$

$$\mathrm{Var}[Z] = (\mu_1 - \mu_2)^\top \big(\frac{\sigma^2}{N\mu^2}I + \alpha_r^2 \frac{\sigma^2}{N_2\mu^2}I\big)(\mu_1 - \mu_2) \tag{F.13}$$

$$= 2\big(\frac{1}{N} + \frac{\alpha_r^2}{N_2}\big)\sigma^2(1 - \cos\angle(\mu_1, \mu_2))$$

Since $Z \sim \mathcal{N}(\mathbb{E}[Z], \mathrm{Var}(Z))$, the probability that the update direction is driven by the majority class is:

$$\Pi_{\mathrm{maj}} < P(Z > 0)$$

$$= \Phi\left(\frac{\mathbb{E}[Z]}{\sqrt{\mathrm{Var}(Z)}}\right)$$

$$= \Phi\left(\frac{\mu(1 - \alpha_r)(1 - \cos\angle(\mu_1, \mu_2))}{\sqrt{2(\frac{1}{N} + \frac{\alpha_r^2}{N_2})\sigma^2(1 - \cos\angle(\mu_1, \mu_2))}}\right) \tag{F.14}$$

$$< \Phi\left(\frac{(1 - \alpha_r)\mu\sqrt{2N(1 - \cos\angle(\mu_1, \mu_2))}}{2\sigma}\right)$$

$$\square$$

## F.4 Multi-class Extensions of Theorem 2.1 and Theorem 3.1

**Theorem F.1** (Multi-class Extension of Theorem 2.1). *Consider a training set with $K$ classes, $\mathcal{Y} = \{1, 2, \ldots, K\}$, and per-class sample counts $N_1 > N_2 > \ldots > N_K$. The expected proportion of parameter matrices where the minority class $K$ is overwhelmed by the majority class $1$ is*

$$\Pi = \Phi\left(\frac{(N_1 - N_K)\mu^2\big(1 - \cos\angle(\mu_1, \mu_K)\big) + \big(\sum_{k=2}^{K-1} N_k\mu_k\big) \cdot (\mu_1 - \mu_K)}{\sqrt{2N\sigma^2\mu^2\big(1 - \cos\angle(\mu_1, \mu_K)\big)}}\right), \tag{F.15}$$

*where $N = \sum_{k=1}^{K} N_k$.*

Here, $(N_1 - N_K) > 0$, and the cross-term $\big(\sum_{k=2}^{K-1} N_k\mu_k\big) \cdot (\mu_1 - \mu_K)$ depends on inter-class similarity. By modeling this inner product as a zero-mean random variable, or by assuming mutually orthogonal classes, the bound simplifies to

$$\Pi = \Phi\left(\frac{(N_1 - N_K)\mu\sqrt{1 - \cos\angle(\mu_1, \mu_K)}}{\sqrt{2N}\,\sigma}\right). \tag{F.16}$$

This highlights the impact of the low-rank component under severe imbalance, mirroring the structural behavior in the two-class setting.

**Theorem F.2** (Multi-class Extension of Theorem 3.1). *Consider the same $K$-class setup. Suppose class $y = K$ contributes gradients to both $W_g$ and $W_e$, while other classes only contribute to $W_g$.*

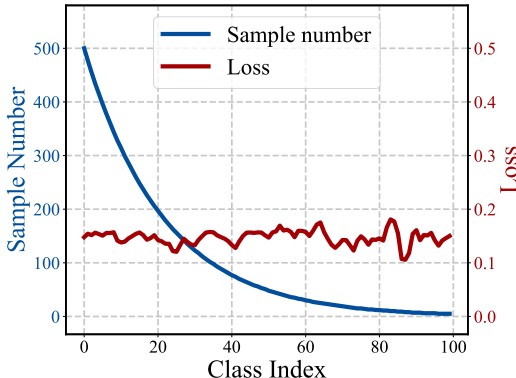

Figure G.1: Per-class raw losses. The setting is same as in Fig. 1(b)

*Then the expected proportion of parameter matrices where class $K$ is overwhelmed by class $1$ is bounded by*

$$\Pi < \Phi\left(\frac{(1-\alpha_r)\mu\sqrt{2N\left(1-\cos\angle(\mu_1,\mu_K)\right)}}{2\sigma}\right). \tag{F.17}$$

Here, $N = \sum_{k=1}^{K} N_k$. We deliberately omit the dependence on class-wise sample numbers in order to more clearly isolate the effect of the low-rank component. This demonstrates that the imbalance-mitigating effect also extends to the multi-class setting.

## G  EXPERIMENTAL SUPPLEMENT

### G.1  IMPLEMENTATION DETAILS

Following Ho et al. (2020), we utilize a U-Net (Ronneberger et al., 2015) based on a Wide ResNet (Zagoruyko and Komodakis, 2016) as the noise prediction network. We set the hyperparameters for DDPM as $\beta_1 = 10^{-4}$ and $\beta_T = 0.02$, with maximum timestep $T = 1000$. The Adam optimizer (Kingma and Ba, 2015) is used with betas $= (0.9, 0.999)$ and a learning rate of $2 \times 10^{-4}$. The dropout rate is set to $0.1$. We use a batch size of $64$ and train the model for $300,000$ steps, including a warm-up period of $5,000$ steps. For the rank of $BA$ in Eq. (1), we fix it at $\frac{1}{10}\min(d,k)$. We only apply the low-rank decomposition to the upsampling part of the U-Net, i.e., the latter half of the model, as the shallow layers tend to capture more general knowledge (Alzubaidi et al., 2021). For the hyperparameter $\lambda$ in Eq. (4), we fix it as $\lambda = 1$. For the base loss in Eq. (4), we adopt the objective function from Zhang et al. (2024), unless otherwise specified. During inference, new images are generated utilizing the 50-step DDIM solver (Song et al., 2021a).

### G.2  MORE RESULTS ABOUT FIG. 1(B)

We provide the explicit definition of the metric used in Fig. 1(b). To measure the sensitivity of different classes to capacity reduction, we calculate the **Relative Loss Change**. Specifically, we identify the top 10% of model parameters with the smallest L1-norms and set them to zero. Let $\mathcal{L}_{raw}^c$ be the original training loss for class $c$, and $\mathcal{L}_{pruned}^c$ be the loss after pruning. The metric is defined as:

$$\text{Relative Loss Change}^c = \frac{\mathcal{L}_{pruned}^c - \mathcal{L}_{raw}^c}{\mathcal{L}_{raw}^c} \tag{G.1}$$

This metric highlights that minority classes are significantly more sensitive to parameter pruning, suggesting they rely on "fragile" parameters.

In addition, we showcase the raw losses and absolute per-class loss changes in Figs. G.1 and G.2.

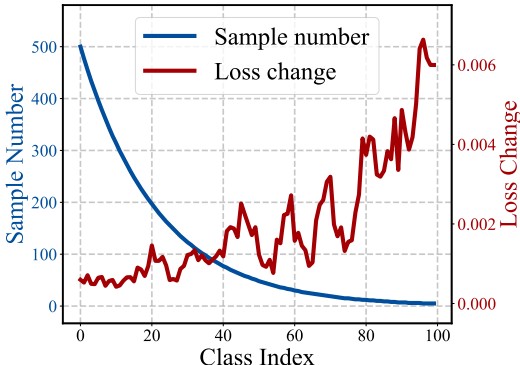

Figure G.2: Per-class absolute loss changes. The setting is same as in Fig. 1(b)

### G.3 STANDARD DEVIATIONS

Due to space limitations, we do not include the standard deviations for Tables 3 and 7 in the main text. The complete results with standard deviations are provided in Tables G.4 to G.6.

Table G.1: FIDs (↓), KIDs (↓), Recalls (↑), and ISs (↑) of CM and various baseline methods on Imb. CIFAR-10 and Imb. CIFAR-100 with different imbalance ratios IR = {100, 50}. All results are reported as Mean ± Std. **Best** and second-best results are highlighted.

| Dataset | Method | FID ↓ | KID ↓ | Recall ↑ | IS ↑ |
|---|---|---|---|---|---|
| Imb. CIFAR-10 IR = 100 | DDPM (Ho et al., 2020) | $10.697 \pm 0.079$ | $0.0035 \pm 0.0008$ | $0.47 \pm 0.01$ | $9.39 \pm 0.12$ |
| | +ADA (Karras et al., 2020) | $9.266 \pm 0.133$ | $0.0029 \pm 0.0003$ | $0.49 \pm 0.02$ | $9.26 \pm 0.14$ |
| | +RS (Mahajan et al., 2018) | $12.332 \pm 0.064$ | $0.0037 \pm 0.0003$ | $0.45 \pm 0.02$ | $9.25 \pm 0.08$ |
| | +Focal (Lin et al., 2017b) | $10.842 \pm 0.134$ | $0.0034 \pm 0.0001$ | $0.46 \pm 0.03$ | $9.42 \pm 0.18$ |
| | CBDM (Qin et al., 2023) | $\underline{8.233 \pm 0.152}$ | $\underline{0.0026 \pm 0.0001}$ | $\mathbf{0.53 \pm 0.02}$ | $9.23 \pm 0.11$ |
| | OC (Zhang et al., 2024) | $8.390 \pm 0.063$ | $0.0027 \pm 0.0002$ | $\underline{0.52 \pm 0.03}$ | $\mathbf{9.53 \pm 0.12}$ |
| | CM | $\mathbf{7.727 \pm 0.124}$ | $\mathbf{0.0023 \pm 0.0001}$ | $\mathbf{0.53 \pm 0.01}$ | $\underline{9.52 \pm 0.10}$ |
| Imb. CIFAR-10 IR = 50 | DDPM (Ho et al., 2020) | $10.216 \pm 0.138$ | $0.0035 \pm 0.0002$ | $0.47 \pm 0.03$ | $9.37 \pm 0.13$ |
| | +ADA (Karras et al., 2020) | $9.132 \pm 0.215$ | $0.0030 \pm 0.0002$ | $0.51 \pm 0.04$ | $9.28 \pm 0.21$ |
| | +RS (Mahajan et al., 2018) | $11.231 \pm 0.177$ | $0.0038 \pm 0.0002$ | $0.47 \pm 0.02$ | $9.31 \pm 0.14$ |
| | +Focal (Lin et al., 2017b) | $10.315 \pm 0.263$ | $0.0034 \pm 0.0003$ | $0.48 \pm 0.01$ | $9.38 \pm 0.23$ |
| | CBDM (Qin et al., 2023) | $\underline{7.933 \pm 0.082}$ | $\underline{0.0026 \pm 0.0002}$ | $\mathbf{0.54 \pm 0.02}$ | $9.42 \pm 0.14$ |
| | OC (Zhang et al., 2024) | $8.034 \pm 0.225$ | $0.0027 \pm 0.0001$ | $\underline{0.53 \pm 0.01}$ | $\underline{9.65 \pm 0.09}$ |
| | CM | $\mathbf{7.372 \pm 0.125}$ | $\mathbf{0.0024 \pm 0.0002}$ | $\mathbf{0.54 \pm 0.01}$ | $\mathbf{9.69 \pm 0.09}$ |
| Imb. CIFAR-100 IR = 100 | DDPM (Ho et al., 2020) | $10.163 \pm 0.077$ | $0.0029 \pm 0.0005$ | $0.46 \pm 0.01$ | $\mathbf{13.45 \pm 0.15}$ |
| | +ADA (Karras et al., 2020) | $9.482 \pm 0.125$ | $0.0032 \pm 0.0002$ | $0.51 \pm 0.01$ | $12.44 \pm 0.16$ |
| | +RS (Mahajan et al., 2018) | $11.432 \pm 0.287$ | $0.0038 \pm 0.0007$ | $0.44 \pm 0.03$ | $12.12 \pm 0.18$ |
| | +Focal (Lin et al., 2017b) | $10.212 \pm 0.110$ | $0.0032 \pm 0.0004$ | $0.47 \pm 0.02$ | $13.07 \pm 0.26$ |
| | CBDM (Qin et al., 2023) | $10.051 \pm 0.391$ | $0.0036 \pm 0.0003$ | $\underline{0.51 \pm 0.01}$ | $12.35 \pm 0.12$ |
| | OC (Zhang et al., 2024) | $\underline{8.309 \pm 0.233}$ | $\underline{0.0026 \pm 0.0002}$ | $\mathbf{0.52 \pm 0.02}$ | $\underline{13.44 \pm 0.20}$ |
| | CM | $\mathbf{7.519 \pm 0.132}$ | $\mathbf{0.0017 \pm 0.0003}$ | $\mathbf{0.52 \pm 0.02}$ | $\mathbf{13.45 \pm 0.23}$ |
| Imb. CIFAR-100 IR = 50 | DDPM (Ho et al., 2020) | $9.363 \pm 0.069$ | $0.0032 \pm 0.0002$ | $0.47 \pm 0.02$ | $\mathbf{14.27 \pm 0.22}$ |
| | +ADA (Karras et al., 2020) | $8.927 \pm 0.138$ | $0.0033 \pm 0.0001$ | $0.51 \pm 0.02$ | $12.89 \pm 0.17$ |
| | +RS (Mahajan et al., 2018) | $10.259 \pm 0.217$ | $0.0037 \pm 0.0003$ | $0.47 \pm 0.03$ | $12.38 \pm 0.23$ |
| | +Focal (Lin et al., 2017b) | $9.477 \pm 0.114$ | $0.0034 \pm 0.0002$ | $0.49 \pm 0.03$ | $13.31 \pm 0.15$ |
| | CBDM (Qin et al., 2023) | $8.946 \pm 0.178$ | $0.0036 \pm 0.0003$ | $\mathbf{0.55 \pm 0.02}$ | $12.59 \pm 0.19$ |
| | OC (Zhang et al., 2024) | $\underline{7.188 \pm 0.274}$ | $\underline{0.0024 \pm 0.0002}$ | $\underline{0.54 \pm 0.01}$ | $13.99 \pm 0.22$ |
| | CM | $\mathbf{6.732 \pm 0.052}$ | $\mathbf{0.0021 \pm 0.0001}$ | $\mathbf{0.55 \pm 0.03}$ | $\underline{14.12 \pm 0.15}$ |

Table G.2: FIDs ($\downarrow$), KIDs ($\downarrow$), and per-class FIDs ($\downarrow$) of CM and baselines on Imb. CelebA-HQ with imbalance ratios IR $= \{100, 50\}$.

| Dataset | Method | Female FID $\downarrow$ | Male FID $\downarrow$ | Overall FID $\downarrow$ | KID $\downarrow$ |
|---|---|---|---|---|---|
| Imb. CelebA-HQ IR $= 100$ | DDPM (Ho et al., 2020) | $7.143 \pm 0.147$ | $16.425 \pm 0.032$ | $8.727 \pm 0.126$ | $0.0037 \pm 0.0001$ |
| | CBDM (Qin et al., 2023) | $\underline{7.043 \pm 0.079}$ | $14.273 \pm 0.183$ | $\underline{7.823 \pm 0.115}$ | $0.0043 \pm 0.0002$ |
| | OC (Zhang et al., 2024) | $7.092 \pm 0.323$ | $\underline{13.962 \pm 0.221}$ | $7.871 \pm 0.237$ | $\underline{0.0034 \pm 0.0002}$ |
| | CM | $\mathbf{6.815 \pm 0.241}$ | $\mathbf{12.788 \pm 0.316}$ | $\mathbf{7.538 \pm 0.201}$ | $\mathbf{0.0033 \pm 0.0002}$ |
| Imb. CelebA-HQ IR $= 50$ | DDPM (Ho et al., 2020) | $7.348 \pm 0.219$ | $14.808 \pm 0.152$ | $8.007 \pm 0.265$ | $\underline{0.0034 \pm 0.0002}$ |
| | CBDM (Qin et al., 2023) | $7.317 \pm 0.273$ | $\underline{12.592 \pm 0.181}$ | $\underline{7.423 \pm 0.139}$ | $0.0042 \pm 0.0001$ |
| | OC (Zhang et al., 2024) | $\underline{7.283 \pm 0.226}$ | $12.938 \pm 0.277$ | $7.438 \pm 0.247$ | $\underline{0.0034 \pm 0.0003}$ |
| | CM | $\mathbf{7.147 \pm 0.182}$ | $\mathbf{11.273 \pm 0.146}$ | $\mathbf{7.193 \pm 0.282}$ | $\mathbf{0.0033 \pm 0.0002}$ |

Table G.3: FIDs ($\downarrow$), KIDs ($\downarrow$), Recalls ($\uparrow$), and ISs ($\uparrow$) of CM and various baseline methods on Imb. ArtBench-10 with imbalance ratios IR $= \{100, 50\}$ using LoRA to fine-tune Stable Diffusion.

| Dataset | Method | FID $\downarrow$ | KID $\downarrow$ | Recall $\uparrow$ | IS $\uparrow$ |
|---|---|---|---|---|---|
| Imb. ArtBench-10 IR $= 100$ | DDPM (Ho et al., 2020) | $27.083 \pm 0.438$ | $0.0142 \pm 0.0003$ | $0.39 \pm 0.01$ | $8.47 \pm 0.19$ |
| | CBDM (Qin et al., 2023) | $25.723 \pm 0.263$ | $0.0122 \pm 0.0002$ | $\underline{0.43 \pm 0.01}$ | $7.97 \pm 0.22$ |
| | OC (Zhang et al., 2024) | $\underline{24.315 \pm 0.162}$ | $\underline{0.0106 \pm 0.0005}$ | $0.42 \pm 0.01$ | $\mathbf{8.71 \pm 0.20}$ |
| | CM | $\mathbf{22.776 \pm 0.078}$ | $\mathbf{0.0087 \pm 0.0002}$ | $\mathbf{00.44 \pm 0.02}$ | $\mathbf{8.71 \pm 0.18}$ |
| Imb. ArtBench-10 IR $= 50$ | DDPM (Ho et al., 2020) | $25.557 \pm 0.082$ | $0.0134 \pm 0.0004$ | $0.39 \pm 0.02$ | $8.41 \pm 0.15$ |
| | CBDM (Qin et al., 2023) | $24.487 \pm 0.153$ | $0.0114 \pm 0.0002$ | $\underline{0.43 \pm 0.02}$ | $8.03 \pm 0.23$ |
| | OC (Zhang et al., 2024) | $\underline{23.287 \pm 0.232}$ | $\underline{0.0097 \pm 0.0003}$ | $\underline{0.43 \pm 0.02}$ | $8.48 \pm 0.17$ |
| | CM | $\mathbf{21.733 \pm 0.153}$ | $\mathbf{0.0080 \pm 0.0002}$ | $\mathbf{0.44 \pm 0.01}$ | $\mathbf{8.51 \pm 0.23}$ |

Table G.5: Per-split FIDs and overall FIDs ($\downarrow$, Mean $\pm$ Std) of DDPM, CM ($\theta^g$), and CM on Imb. CIFAR-100 with imbalance ratio IR $= 100$. Many, Medium, and Few are the three splits based on the training imbalance. **Best** results are highlighted.

| Dataset | Method | Many FID $\downarrow$ | Med. FID $\downarrow$ | Few FID $\downarrow$ | Overall FID $\downarrow$ |
|---|---|---|---|---|---|
| Imb. CIFAR-100 IR $= 100$ | DDPM (Ho et al., 2020) | $14.068 \pm 0.193$ | $15.660 \pm 0.047$ | $22.188 \pm 0.241$ | $10.163 \pm 0.077$ |
| | CM ($\theta^g$) | $11.923 \pm 0.139$ | $14.872 \pm 0.157$ | $29.357 \pm 0.318$ | $13.712 \pm 0.240$ |
| | CM ($\theta = \theta^g \oplus \theta^e$) | $\mathbf{11.713 \pm 0.247}$ | $\mathbf{13.043 \pm 0.138}$ | $\mathbf{18.729 \pm 0.141}$ | $\mathbf{7.519 \pm 0.132}$ |

Table G.6: FIDs ($\downarrow$), KIDs ($\downarrow$), Recalls ($\uparrow$), and ISs ($\uparrow$) on Imb. CIFAR-100 with imbalance ratio IR $= 100$. The last two rows show the results of CM after removing $\mathcal{L}_{\mathrm{Con}}$ and $\mathcal{L}_{\mathrm{Div}}$, respectively. **Best** and second-best results are highlighted.

| Dataset | Method | FID $\downarrow$ | KID $\downarrow$ | Recall $\uparrow$ | IS $\uparrow$ |
|---|---|---|---|---|---|
| Imb. CIFAR-100 IR $= 100$ | DDPM (Ho et al., 2020) | $10.163 \pm 0.077$ | $0.0029 \pm 0.0005$ | $0.46 \pm 0.01$ | $\mathbf{13.45 \pm 0.15}$ |
| | OC (Zhang et al., 2024) | $8.309 \pm 0.233$ | $0.0026 \pm 0.0002$ | $\mathbf{0.52 \pm 0.02}$ | $\underline{13.44 \pm 0.20}$ |
| | CM | $\mathbf{7.519 \pm 0.132}$ | $\mathbf{0.0017 \pm 0.0003}$ | $\mathbf{0.52 \pm 0.02}$ | $\mathbf{13.45 \pm 0.23}$ |
| | CM w/o $\mathcal{L}_{\mathrm{Con}}$ | $8.412 \pm 0.227$ | $0.0029 \pm 0.0002$ | $0.50 \pm 0.01$ | $13.23 \pm 0.22$ |
| | CM w/o $\mathcal{L}_{\mathrm{Div}}$ | $\underline{8.073 \pm 0.174}$ | $\underline{0.0025 \pm 0.0001}$ | $\underline{0.51 \pm 0.01}$ | $13.42 \pm 0.16$ |

## G.4 HUMAN PERCEPTUAL STUDY

We conducted a user study focusing on the minority classes of Imb. ArtBench-10 (IR=100). Eight human evaluators participated, each assessing between 20 and 36 image groups. In each group, we provided 10 real images (explicitly identified) together with 2 generated images: one produced by the baseline (DDPM) and the other by our method CM, presented in random order. All images within a group belonged to the same class. Evaluators were asked to select the generated image that

Table G.4: Per-split FIDs (↓) of CM and baselines on Imb. CIFAR-10 (IR = 100) and Imb. CIFAR-100 (IR = 100). Many, Medium, and Few are the three splits based on the training imbalancedness. **Best** and second-best results are highlighted.

| Imb. CIFAR-10, IR = 100 | | | | | |
|---|---|---|---|---|---|
| Dataset | Method | Many FID ↓ | Med. FID ↓ | Few FID ↓ | Overall FID ↓ |
| Imb. CIFAR-10 IR = 100 | DDPM (Ho et al., 2020) | 14.203 ± 0.221 | 19.714 ± 0.075 | 15.869 ± 0.152 | 10.697 ± 0.079 |
| | CBDM (Qin et al., 2023) | 12.222 ± 0.087 | 14.465 ± 0.072 | 12.230 ± 0.107 | 8.2334 ± 0.152 |
| | OC (Zhang et al., 2024) | 12.026 ± 0.102 | 15.234 ± 0.108 | 12.254 ± 0.178 | 8.3896 ± 0.063 |
| | CM | **11.705 ± 0.105** | **14.340 ± 0.093** | **11.218 ± 0.101** | **7.7273 ± 0.124** |
| Imb. CIFAR-100 IR = 100 | DDPM (Ho et al., 2020) | 14.068 ± 0.193 | 15.660 ± 0.047 | 22.188 ± 0.241 | 10.163 ± 0.077 |
| | CBDM (Qin et al., 2023) | 12.585 ± 0.182 | 14.042 ± 0.273 | 20.667 ± 0.294 | 10.051 ± 0.391 |
| | OC (Zhang et al., 2024) | 11.731 ± 0.221 | 13.133 ± 0.073 | 21.053 ± 0.371 | 8.309 ± 0.233 |
| | CM | **11.713 ± 0.247** | **13.043 ± 0.138** | **18.729 ± 0.141** | **7.519 ± 0.132** |

Table G.7: User study results on minority classes of Imb. ArtBench-10 (IR=100). Eight evaluators compared generated images from the baseline (DDPM) and our method CM. Numbers indicate the count of trials where each method was preferred.

| Method | Eval. 1 | Eval. 2 | Eval. 3 | Eval. 4 | Eval. 5 | Eval. 6 | Eval. 7 | Eval. 8 | Overall |
|---|---|---|---|---|---|---|---|---|---|
| DDPM | 10 | 10 | 4 | 4 | 6 | 3 | 14 | 12 | **63** |
| CM | 24 | 26 | 16 | 19 | 18 | 17 | 15 | 16 | **151** |

demonstrated higher realism and stylistic fidelity. As shown in Tab. G.7, our method was preferred in 70.56% of the trials, indicating a clear perceptual advantage over the baseline.

## G.5 DOWNSTREAM UTILITY ASSESSMENT

To evaluate the utility of generated samples, we augmented the Imb. CIFAR-100 (IR=100) dataset for a downstream classification task. A ResNet-18 classifier was trained under three settings: (i) no augmentation, (ii) augmentation with 50k images generated by the baseline (DDPM), and (iii) augmentation with 50k images generated by our method (CM). As shown in Tab. G.8, augmentation with CM-generated images yields a significantly stronger downstream classifier, demonstrating the superior quality and diversity of our minority-class generations.

## G.6 MORE ABLATIONS

In Fig. G.3, we present additional ablation studies on the hyperparameter $\lambda$ across more datasets.

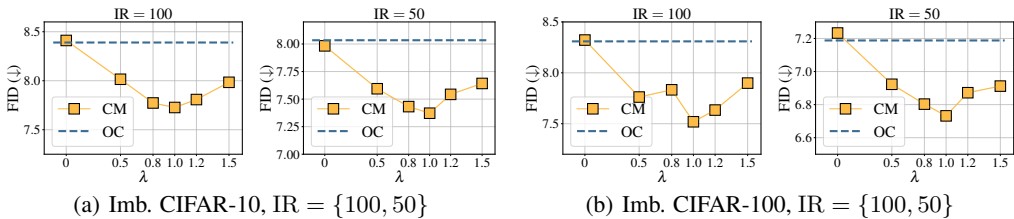

(a) Imb. CIFAR-10, IR = {100, 50}                    (b) Imb. CIFAR-100, IR = {100, 50}

Figure G.3: Ablation study on the hyperparameter $\lambda$ in Eq. (4). We use OC as a reference because it shows the best overall performance among the baselines and serves as our base loss. Figures (a) and (b) show results on Imb. CIFAR-10 and Imb. CIFAR-100, respectively, with imbalance ratios of IR = 100 and IR = 50 from left to right. We report FIDs for $\lambda = \{0.0, 0.5, 0.8, 1.0, 1.2, 1.5\}$.

Table G.8: Downstream classification performance on Imb. CIFAR-100 (IR=100) using ResNet-18. Augmentation with images generated by CM leads to clear improvements over both the baseline (DDPM) and the no-augmentation setting.

| Method | Precision | Recall |
|---|---|---|
| No augmentation | 43.25 | 37.25 |
| w/ DDPM 50k images | 46.73 | 42.34 |
| w/ CM 50k images | **49.23** | **46.78** |

## G.7 VISUAL ANALYSIS OF CAPACITY ALLOCATION

To visually validate the roles of the decomposed parameters, we conducted a qualitative ablation study on Imb. ArtBench-10 (IR = 100). We generated samples for minority classes using only the general branch ($\theta^g$) and compared them with the full model ($\theta = \theta^g \oplus \theta^e$). As shown in Fig. G.4, samples generated by $\theta^g$ effectively capture the high-level semantic structure and generic features (e.g., basic object shapes and backgrounds) dominated by majority classes. However, they fail to render the specific artistic styles and fine-grained details unique to the minority classes. In contrast, the full model successfully recovers these minority-specific traits. This empirically confirms that our Capacity Manipulation effectively disentangles feature learning: generic knowledge is stored in $\theta^g$, while minority expertise is preserved in $\theta^e$.

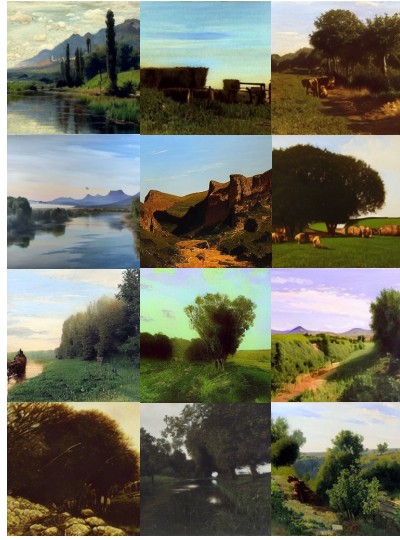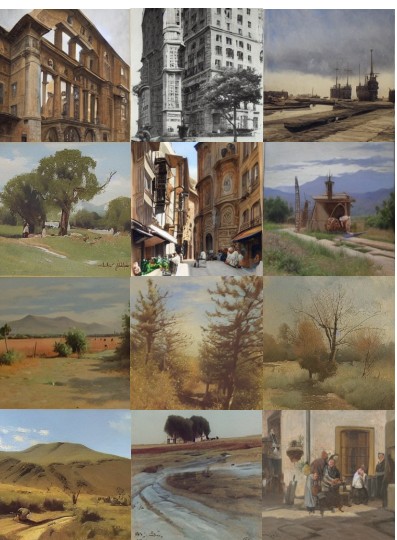

Figure G.4: Visual comparison between using only generic parameters ($\theta^g$, left) and the full model ($\theta^g \oplus \theta^e$, right) on the minority class (Realism) of Imb. ArtBench-10. $\theta^g$ captures generic features but misses minority-specific details, which are restored by $\theta^e$.

## G.8 MORE VISUALIZATION

The generated images for one of the medium classes "surrealism" on Imb. ArtBench-10 with IR = 100 are shown in Fig. G.7. It is evident that the generated styles of CM are much closer to the real images. The generated images for the minority class "Male" on Imb. CelebA-HQ with IR = 100 are shown in Fig. G.6. More visualization of generation results with CM are presented in Fig. G.8 (Imb. CIFAR-100, IR = 100), Fig. G.9 (Imb. CelebA-HQ, IR = 100), Fig. G.11 (ImageNet-LT), Fig. G.12 (iNaturalist), and Fig. G.10 (Imb. ArtBench-10, IR = 100).

**Diversity in the Tail.** To further demonstrate that our method avoids mode collapse in minority classes, we provide additional visualizations focusing on the intra-class diversity of the tail. As shown in Fig. G.5, our CM generates minority samples with significant variations, closely mirroring

the diversity of the real data distribution. This aligns with our high Recall scores reported in the main experiments.

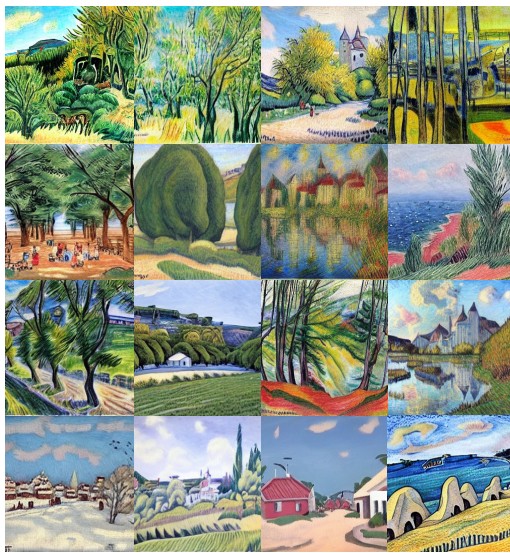 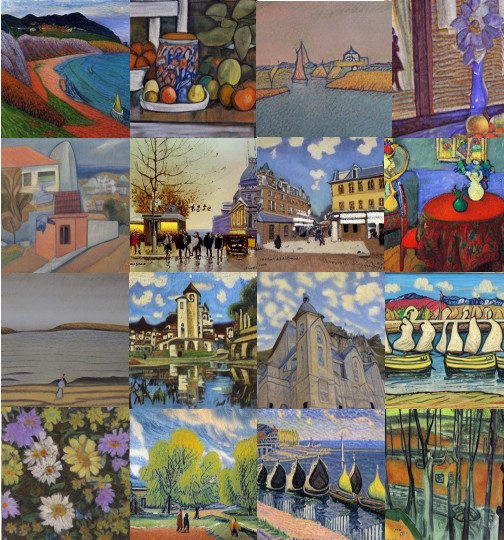

Figure G.5: Additional visualization of intra-class diversity for the minority class (Post Impressionism, Imb. ArtBench-10, IR = 100). (Left) DDPM. (Right) CM. The generated samples exhibit diverse poses, backgrounds, and styles, confirming that CM effectively mitigates mode collapse in the tail.

Imb. CelebA-HQ, IR = 100, Male

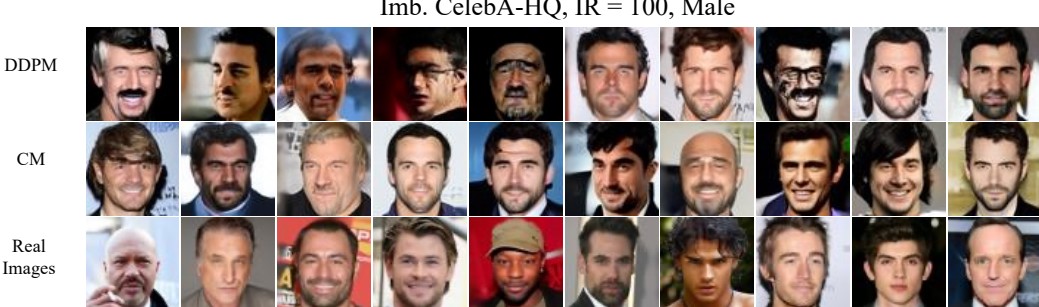

Figure G.6: The visualization of generated images on Imb. CelebA-HQ with imbalance ratio IR = 100. The figure showcases the generated outputs for the class "Male", which is the minority class, from both DDPM and CM. The last row displays real images from the dataset for reference. It is evident that CM generates more realistic and diverse faces.

Imb. ArtBench-10, IR = 100, Surrealism

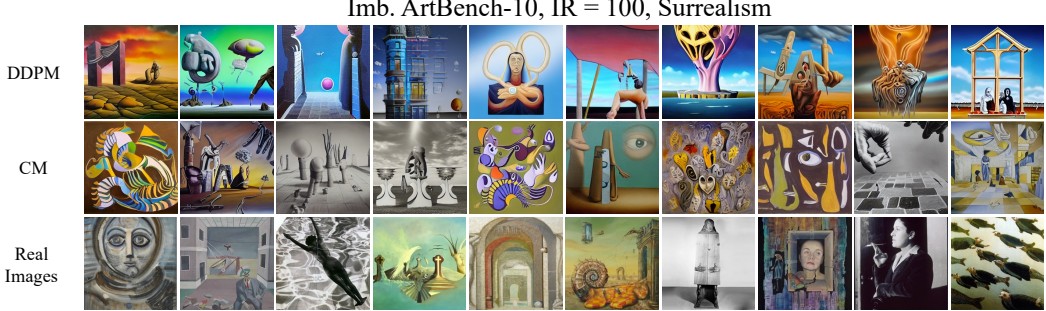

Figure G.7: The visualization of generated images on Imb. ArtBench-10 with imbalance ratio IR = 100. The figure showcases the generated outputs for the class "Surrealism", which is one of the medium classes, from both DDPM and CM. The last row displays real images from the dataset for reference. It is evident that CM generates results that are significantly more stylistically closer to the real images compared to DDPM. The images shown are randomly selected.

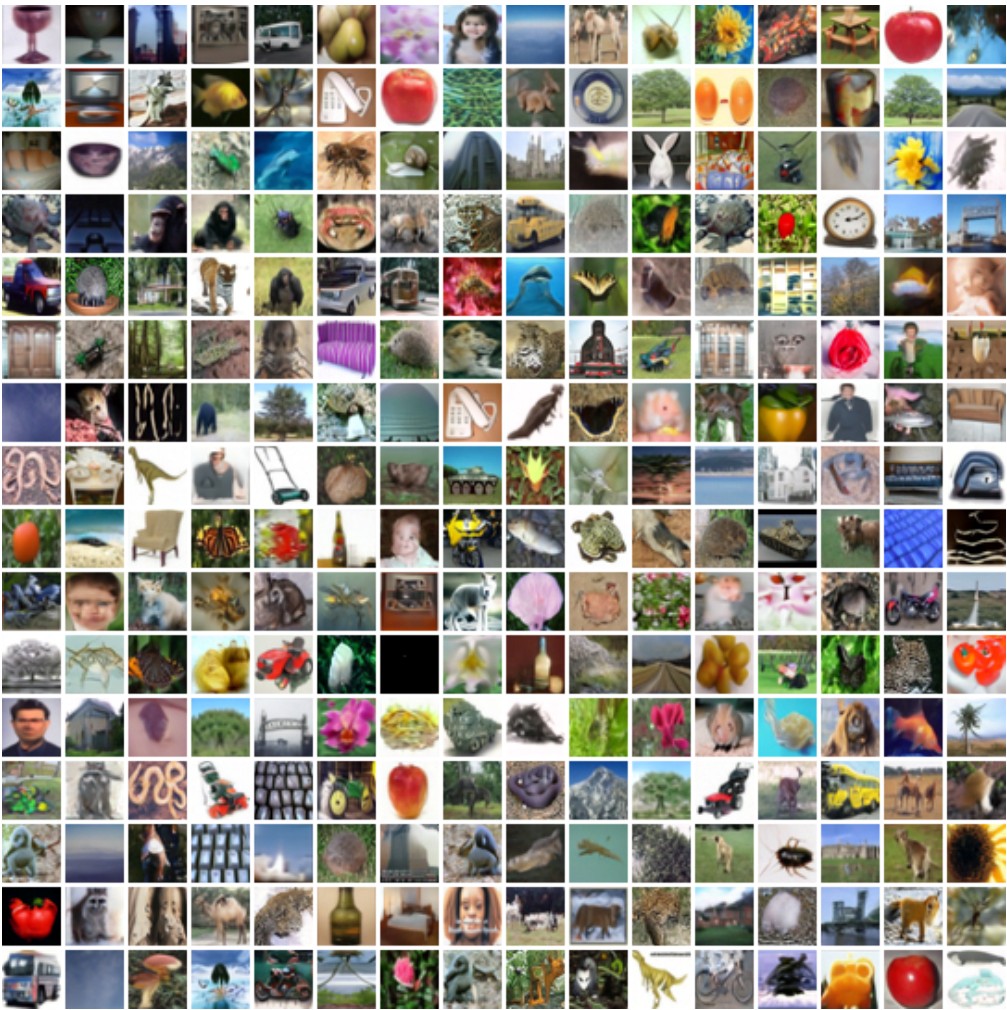

Figure G.8: Visualization of generation results on Imb. CIFAR-100 (IR = 100) with CM. The images shown are randomly selected.

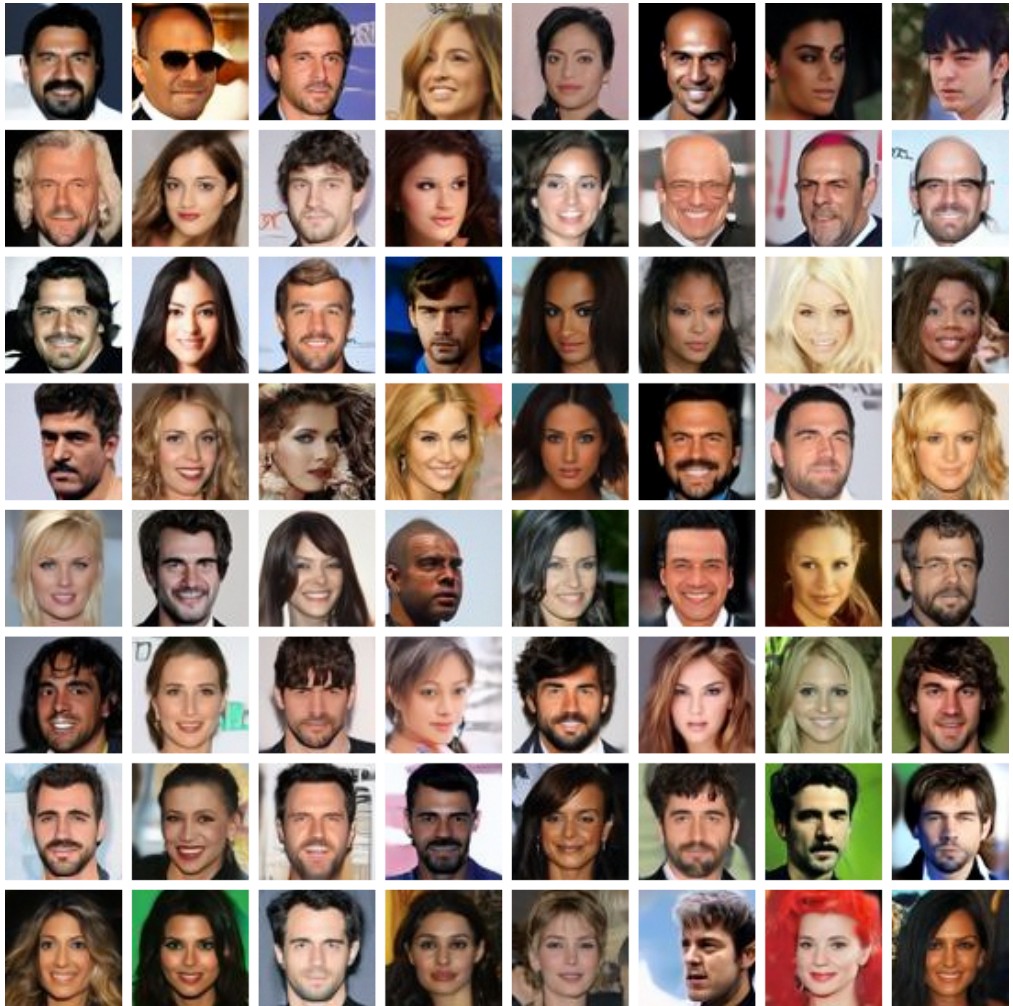

Figure G.9: Visualization of generation results on Imb. CelebA-HQ (IR = 100) with CM. The images shown are randomly selected.

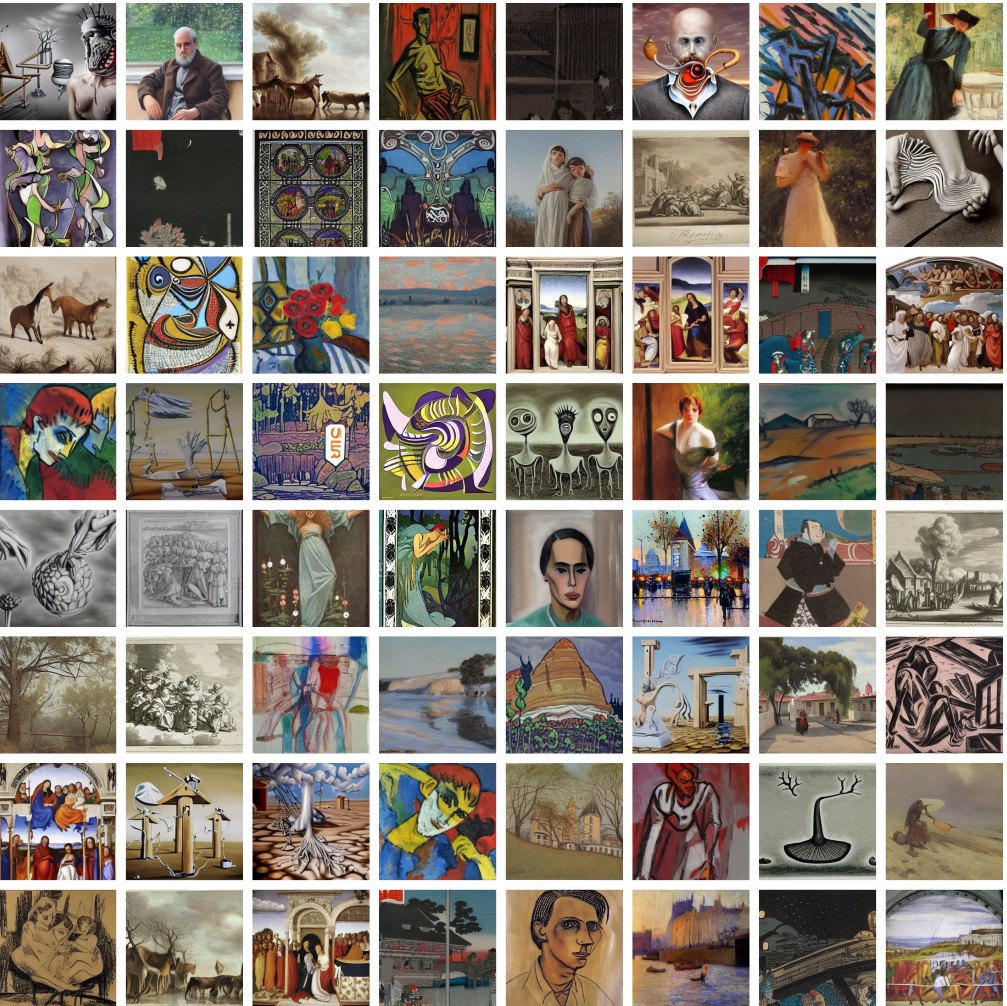

Figure G.10: Visualization of generation results on Imb. ArtBench-10 (IR = 100) with CM. The images shown are randomly selected.

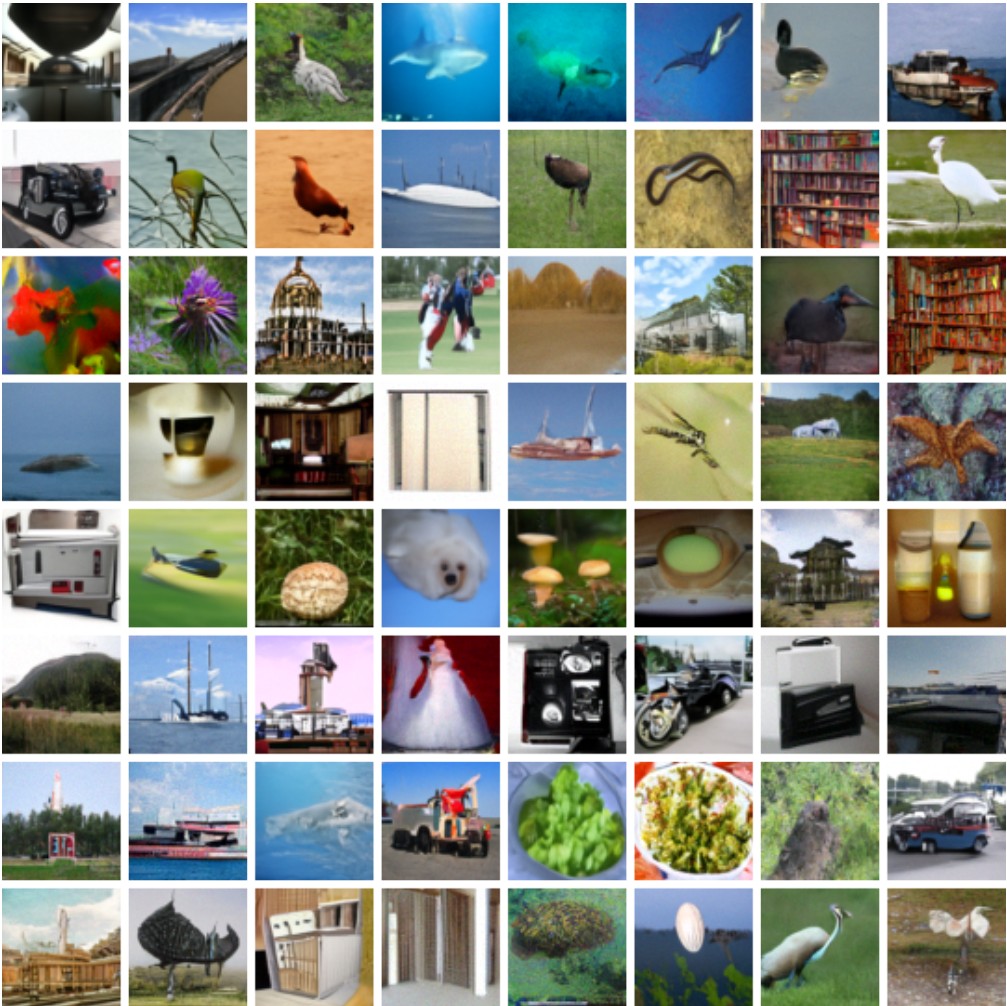

Figure G.11: Visualization of generation results on ImageNet-LT with CM. The images shown are randomly selected.

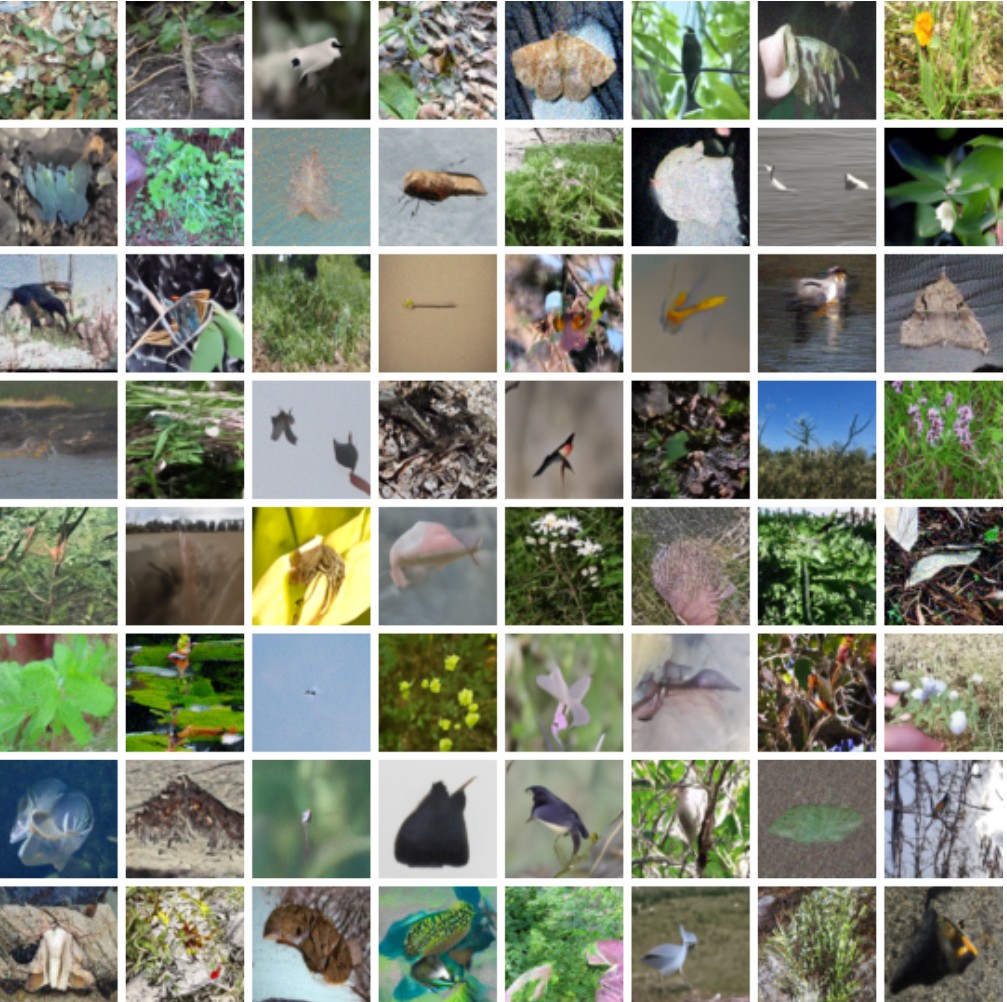

Figure G.12: Visualization of generation results on iNaturalist with CM. The images shown are randomly selected.

Table G.9: FID results on Consistency Models. Incorporating our method (+CM) consistently reduces FID compared to the baseline.

| Dataset (IR=100) | Baseline FID | +CM FID |
|---|---|---|
| Imbalanced CIFAR-10 | 10.523 | **9.257** |
| Imbalanced CIFAR-100 | 9.862 | **8.638** |

Table G.10: FID results on DPM-Solver (15 steps). Our method (+CM) improves generation quality over the baseline.

| Dataset (IR=100) | Baseline FID | +CM FID |
|---|---|---|
| Imbalanced CIFAR-10 | 10.932 | **7.623** |
| Imbalanced CIFAR-100 | 11.524 | **8.033** |

Table G.11: FID results on latent diffusion models trained from scratch on fine-grained text-to-image datasets. Our method (+CM) improves generation quality over the baseline.

| Dataset | Baseline FID | +CM FID |
|---|---|---|
| CUB-200 | 24.62 | **22.64** |
| Oxford-102 | 26.47 | **24.31** |

## G.9 BROADER SETTINGS

**Consistency Models.** Beyond DDPM, we also validated our method on the Consistency Models (Song et al., 2023). As shown in Tab. G.9, incorporating our method CM consistently improves FID scores across datasets.

**Integration with DPM-Solver.** We further validated our method with DPM-Solver (Lu et al., 2022) (15 steps). As shown in Tab. G.10, incorporating our method (+CM) consistently reduces FID across datasets compared to the baseline.

**Latent diffusion models from scratch and fine-grained text-to-image datasets.** We evaluated our method on latent diffusion models trained from scratch using fine-grained text-to-image datasets. Specifically, we conducted experiments on CUB-200 and Oxford-102, creating imbalanced versions by downsampling based on class labels. We followed the network configuration and training details of VQ-Diffusion-S (Gu et al., 2022), using textual descriptions as conditioning for image generation. As shown in Tab. G.11, our method (+CM) consistently improves FID scores over the baseline.

## G.10 COMPARISON WITH OVERLAP OPTIMIZATION

We conduct a direct experimental comparison with Overlap Optimization (Yan et al., 2024) on the Imb. CIFAR-100 (IR = 100) dataset. As shown in Tab. G.12, our method (CM) outperforms Overlap Optimization across key metrics, further validating the effectiveness of our capacity manipulation strategy.

Table G.12: Comparison with Overlap Optimization on Imb. CIFAR-100 (IR = 100).

| Method | FID ↓ | Recall ↑ |
|---|---|---|
| DDPM | 10.16 | 0.46 |
| Overlap Opt. (Yan et al., 2024) | 8.98 | 0.48 |
| **CM (Ours)** | **7.52** | **0.52** |

### G.11 EXTENSION TO DISCRIMINATIVE TASKS

While this paper focuses on diffusion models, the core concept of Capacity Manipulation (CM)—explicitly reserving model capacity for minority classes—is potentially generalizable to discriminative long-tailed recognition tasks. To validate this, we conducted preliminary experiments on Long-Tailed CIFAR-100 with imbalance ratios of $IR = \{200, 100\}$.

**Setup.** We employ ResNet-18 as the backbone. We apply our low-rank decomposition ($\theta = \theta^g \oplus \theta^e$) to the convolutional layers. We compare standard Cross-Entropy (CE), LDAM Loss, and Logit Adjustment (LA) against their CM-enhanced versions.

**Results.** As shown in Tab. G.13, CM yields consistent improvements over the baselines. Notably, at $IR = 100$, CM improves LA by **1.78%** in overall accuracy, with significant gains in Medium and Few classes. This confirms that preventing majority classes from monopolizing the shared feature extractor is beneficial for discriminative tasks as well.

Table G.13: Classification accuracy (%) on Long-Tailed CIFAR-100 using ResNet-18. We compare Cross-Entropy (CE), LDAM, and Logit Adjustment (LA) with their CM-integrated versions under imbalance ratios (IR) of 200 and 100.

| Imbalance Ratio | Method | Many | Med. | Few | Overall |
|---|---|---|---|---|---|
| **IR = 200** | CE | **70.03** | 42.13 | 6.38 | 36.56 |
| | LDAM | 62.13 | 43.64 | 16.64 | 38.66 |
| | **LDAM + CM** | 60.20 | 46.35 | **20.69** | 40.50 |
| | LA | 60.46 | 45.16 | 20.43 | 40.11 |
| | **LA + CM** | 63.73 | **46.38** | 20.16 | **41.36** |
| **IR = 100** | CE | **70.14** | 40.54 | 6.73 | 40.76 |
| | LDAM | 61.77 | 41.57 | 19.10 | 41.90 |
| | **LDAM + CM** | 62.40 | 44.94 | 21.77 | 44.10 |
| | LA | 57.91 | 45.80 | 23.53 | 43.36 |
| | **LA + CM** | 61.60 | **46.94** | **23.80** | **45.14** |

### G.12 ON REPRODUCIBILITY

All the experiments are conducted on NVIDIA A100s and H200s with Python 3.8 and Pytorch 2.0.1. We have provided experimental setups and implementation details in Sec. 4.1 and Sec. 4.2. To ensure reproducibility, the source code will be released upon acceptance.

## H SOCIAL IMPACT

In this paper, we propose a method to enhance the robustness of generative diffusion models against imbalanced data distributions. This advancement holds significant social implications, both positive and negative. On the positive side, our approach could democratize access to high-quality data generation, allowing marginalized communities to benefit from more equitable representation in AI applications. By improving the model's performance on underrepresented classes, we can foster inclusivity in various fields, such as healthcare, finance, and education, where data-driven decisions can impact lives. Conversely, there are potential negative consequences to consider. As generative models become more powerful, they may be misused to create deceptive content, leading to misinformation and erosion of trust in digital media. Additionally, our method's emphasis on underrepresented segments in the training data poses a risk of data poisoning if supervision is lacking. Malicious actors could exploit this focus to introduce biased or harmful data, compromising the model's integrity. This vulnerability underscores the need for robust monitoring and validation mechanisms to ensure data reliability, as any compromise could lead to unintended negative consequences. Therefore, proactive data governance is essential to mitigate these risks while maximizing the benefits of our proposed method.

## I  LIMITATIONS, DISCUSSIONS, AND FUTURE WORK

Our work introduces CM as a novel perspective for addressing class imbalance in diffusion models, demonstrating strong empirical results. However, acknowledging the scope and potential extensions of this work is important for contextualizing our findings and guiding future research. We demonstrated CM's efficacy across various image datasets and both training-from-scratch and fine-tuning scenarios. Evaluating the applicability and potential adaptations of capacity manipulation for different data modalities (*e.g.*, video, 3D data) and other types of generative models remains an interesting area for future exploration.

While CM demonstrates robustness to extreme imbalance ratios (*e.g.*, $\mathrm{IR} = 500$ on iNaturalist), we observe limitations in scenarios of *absolute sample scarcity*. When the absolute number of minority samples is extremely low (*e.g.*, single-digit samples), the reserved capacity $\theta^e$ lacks sufficient data to learn meaningful representations, limiting generation quality despite the explicit capacity allocation. Addressing this "few-shot" to "zero-shot" transition remains a challenging direction for future research.

## J  USE OF LARGE LANGUAGE MODELS (LLMs)

For transparency, we note that large language models (LLMs) (*e.g.*, GPT-5 and Gemini 2.5 Pro) were employed during the preparation of this manuscript. The LLM was used exclusively for language polishing, grammar checking, and formatting. It was not involved in generating research ideas, designing methods, analyzing data, or drawing conclusions. All technical contributions are solely the work of the authors.

