# OpenReview forum: "Improving Diffusion Models for Class-imbalanced Training Data via Capacity Manipulation"
_ICLR.cc/2026/Conference — ICLR 2026 Oral_

### Official Review · Reviewer_JwQN · 2025-10-28

**Soundness:** 3
**Presentation:** 3
**Contribution:** 3
**Rating:** 6
**Confidence:** 4

**Summary:**

This work investigates the challenge of generative modeling for imbalanced datasets. The hypothesis is that the poor generation quality for minority classes is primarily caused by an imbalance in "model capacity," where the model's learning resources are disproportionately occupied by the head (majority) classes. To address this, the paper introduces a novel technique named Capacity Manipulation (CM), which explicitly reallocates and reserves model capacity for the tail (minority) classes. The proposed method employs a low-rank decomposition of the model's parameters, enabling fine-grained control over capacity allocation. A bespoke capacity manipulation loss function is introduced to ensure sufficient capacity is dedicated to learning the features of minority classes, leading to a significant enhancement in their generative representation. The claims are substantiated by comprehensive experimental results, and the overall methodology is presented with clear and coherent logic.

**Strengths:**

1. I find this approach remarkably novel in how it attributes the class imbalance problem to "uneven model capacity allocation." It represents a significant departure from traditional paradigms like data resampling or loss re-weighting, introducing a fresh perspective by intervening directly at the model parameter level. By the way, I'm also curious if the author's method could be applied to long-tail recognition tasks (e.g., with ResNeXt-50 on CIFAR). No detailed explanation is needed if the implementation is complex—I'm simply wondering about its potential.
2. The design of  loss function is exceptionally clear in its objective. By creating a "push-pull" dynamic between 'consistency' and 'diversity', it effectively channels distinct knowledge into separate parameter subspace.
3. The paper doesn't just rest on solid experimental results; it also provides theoretical analysis (Theorems 2.1 and 3.1) to substantiate its core thesis: that majority classes indeed dominate parameter updates and that low-rank decomposition can effectively mitigate this dominance.
4. The experimental validation is remarkably comprehensive. It covers a wide range of datasets (from simple to complex, low-res to high-res), various imbalance ratios, and multiple evaluation metrics, all benchmarked against strong baseline methods.

**Weaknesses:**

1. I'm also curious if the author's method could be applied to long-tail recognition tasks (e.g., with ResNeXt-50 on CIFAR). No detailed explanation is needed if the implementation is complex—I'm simply wondering about its potential.
2. My main question is about the capacity 'reservation.' The structure of the parameter decomposition seems to be fixed. This makes me wonder: is this 'hard partitioning' approach truly optimal? Could there be a way for the model to dynamically and adaptively decide how much capacity to allocate to each component during training, rather than relying on a predefined split?
3. Are there any toy experiments that can visually illustrate this? For instance, using the two-class example you mentioned, could you show how the majority class ends up occupying most of the model's parameter capacity?
4. I think this assumption has some limitations, especially with varying balance ratios like in ImageNet-LT. For example, in an extreme case with one head class and 999 tail classes, is a single, the setting of rank still appropriate? Or does the rank itself need to be adjusted based on class frequency?
5. To empirically validate the hypothesis, is it possible to visually demonstrate that the class-specific parameters specialize in learning features unique to minority classes, while the class-agnostic parameters focus on capturing generic features dominated by the majority (head) classes? We propose achieving this through visualization techniques.
6. My last question is about the diversity within the tail itself. Can you visualize them?

**Questions:**

Please see the weakness.

---

> ### Author Response · Authors · 2025-11-21
> **Response to Reviewer JwQN (1/2)**
>
> > I'm also curious if the author's method could be applied to long-tail recognition tasks (e.g., with ResNeXt-50 on CIFAR). No detailed explanation is needed if the implementation is complex—I'm simply wondering about its potential.
>
> This is an insightful question. We believe the core concept of Capacity Reservation is indeed generalizable to discriminative tasks. To validate this potential, we conducted a preliminary experiment on Long-Tailed CIFAR-100 ($\mathrm{IR}=100$) using a ResNet-18 backbone.
>
> We applied our low-rank decomposition ($\theta = \theta^g \oplus \theta^e$) to the convolutional layers of ResNet-18. As shown below, CM yields a clear improvement.
>
> | Imbalance Ratio | Method       | Many   | Med.   | Few    | Overall    |
> |-----------------|--------------|--------|--------|--------|--------|
> | **200**         | CE           | **70.03**  | 42.13  | 6.38   | 36.56  |
> |                 | LDAM         | 62.13  | 43.64  | 16.64  | 38.66  |
> |                 | **LDAM + CM**  | 60.20  | 46.35  | **20.69**  | 40.50  |
> |                 | LA           | 60.46  | 45.16  | 20.43  | 40.11  |
> |                 | **LA + CM**   | 63.73  | **46.38**  | 20.16  | **41.36**  |
>
> | Imbalance Ratio | Method       | Many   | Med.   | Few    | Overall    |
> |-----|-----|--------|----|--------|---|
> | **100**         | CE           | **70.14**  | 40.54  | 6.73   | 40.76  |
> |                 | LDAM         | 61.77  | 41.57  | 19.10  | 41.90  |
> |                 | **LDAM + CM**  | 62.40  | 44.94  | 21.77  | 44.10  |
> |                 | LA           | 57.91  | 45.80  | 23.53  | 43.36  |
> |                 | **LA + CM**    | 61.60  | **46.94**  | **23.80**  | **45.14**  |
>
> The results confirm that explicitly reserving capacity for minority classes helps prevent the majority classes from dominating the shared feature extractor, thereby improving tail-class recognition.
>
> > My main question is about the capacity 'reservation.' The structure of the parameter decomposition seems to be fixed. This makes me wonder: is this 'hard partitioning' approach truly optimal? Could there be a way for the model to dynamically and adaptively decide how much capacity to allocate to each component during training, rather than relying on a predefined split?
>
> We agree with the reviewer that dynamic or adaptive capacity allocation is a promising direction for future research. However, we believe our current design represents a solid and necessary first step in this direction, striking an practical balance between performance, simplicity, and robustness.
>
> We support this claim with the following points:
> - **Robustness to Partition Size (Rank Ablation):** As shown in Fig. 4(c) of our paper, we conducted an ablation study on the rank of the reserved capacity ($W^e$). The results demonstrate that CM consistently outperforms the state-of-the-art baseline (OC) across a wide range of rank ratios (from 0.05 to 0.25). This indicates that while a "perfect" dynamic split might exist, the model is not overly sensitive to the fixed partition size, and our "hard" reservation is sufficient to yield significant improvements.
> - **Efficiency and Simplicity:** A key advantage of our approach is its simplicity and zero inference overhead. Implementing dynamic allocation (e.g., via routing networks or architectural search) often introduces significant training instability and computational complexity.
> - **Foundational Step:** We position this work as the first to explicitly verify the hypothesis that "capacity monopoly" is a root cause of imbalance. Our fixed decomposition effectively validates this hypothesis and serves as a strong baseline for future adaptive methods.
>
>
> > Are there any toy experiments that can visually illustrate this? For instance, using the two-class example you mentioned, could you show how the majority class ends up occupying most of the model's parameter capacity?
>
> Yes, we can illustrate this phenomenon using a toy experiment on a 2D imbalanced dataset (Imbalance Ratio = 100). We trained a simple MLP and tracked the Cosine Similarity between the model parameter update ($\Delta \theta$) and the expected gradient directions of the majority ($\mathbb{E}[g_{maj}]$) vs. minority ($\mathbb{E}[g_{min}]$) classes.
>
> **Result:** The parameter update vector is almost perfectly aligned with the majority class gradients (Cosine Sim $\approx$ 1.0), confirming our Theorem 2.1 that the majority class dominates the capacity allocation by dictating the optimization trajectory. We tracked the results across all 30 epochs and present a subset of them in the table below.
>
>
>
> | Epoch                                        | 5       | 15      | 25      | 30      | Avg. across 30 epochs |
> | ----- | --- | ------- | --- | ------- | --- |
> | Cosine Similarity (Update vs. Majority Grad) | 0.99967 | 0.99975 | 0.99971 | 0.99966 | 0.99971               |
> | Cosine Similarity (Update vs. Minority Grad) | 0.01457 | 0.01963 | 0.02499 | 0.02790 | 0.019953              |

---

> ### Author Response · Authors · 2025-11-21
> **Response to Reviewer JwQN (2/2)**
>
> > I think this assumption has some limitations, especially with varying balance ratios like in ImageNet-LT. For example, in an extreme case with one head class and 999 tail classes, is a single, the setting of rank still appropriate? Or does the rank itself need to be adjusted based on class frequency?
>
>
> This is an insightful observation. We believe the fixed rank setting remains effective and appropriate for the following reasons, based on both theoretical principles and empirical evidence:
>
> - Principle: Gradient Protection & Shared Representations. The primary role of the reserved capacity ($W^e$) is to provide a protected subspace for minority gradients, preventing them from being overwhelmed by the dominant magnitude of majority updates (as analyzed in Theorem 3.1).
>     - **Shared Knowledge:** In many-class scenarios (like iNaturalist), tail classes often share high-level semantic features. A low-rank $W^e$ effectively captures these shared minority patterns without needing to model each class independently.
>     - **Regularization:** Since tail classes have very few samples, assigning a high rank (large capacity) adaptive to their count could risk **overfitting**. The low-rank constraint acts as a regularizer, forcing the model to learn generalizable features for the tail.
>
> - Empirical: Robustness on Large-Scale Datasets.
>     - Our experiments on ImageNet-LT (1,000 classes) and iNaturalist (8,142 classes) 6demonstrate that the fixed rank setting significantly outperforms baselines even when the number of tail classes is massive.
>     - **Rank Insensitivity:** Furthermore, our ablation study (Fig. 4(c)) shows that CM maintains consistent performance improvements across a wide range of rank ratios8. This suggests that precise rank tuning based on class frequency is not strictly necessary for robustness.
>
>
> - Efficiency Trade-off. While adaptive rank is an interesting direction, a fixed rank allows us to merge weights perfectly during inference, maintaining zero inference overhead. Adaptive ranks would significantly complicate this merging process and increase deployment costs.
>
>
> > To empirically validate the hypothesis, is it possible to visually demonstrate that the class-specific parameters specialize in learning features unique to minority classes, while the class-agnostic parameters focus on capturing generic features dominated by the majority (head) classes? We propose achieving this through visualization techniques.
>
> We agree that visualizing the distinct roles of the decomposed parameters strengthens our hypothesis. We address this through both existing quantitative evidence and a new visual experiment.
>
> - Existing Quantitative Evidence (Table 7): As analyzed in Section 4.3 (Knowledge Allocation), we compared the full model ($\theta = \theta^g \oplus \theta^e$) against a model using only the general parameters ($\theta^g$). The ablation CM ($\theta^g$) maintains strong performance on **Many** classes (FID 11.92) but suffers a severe performance drop on **Few** classes (FID 29.36 vs. 18.73 for Full CM). This quantitatively confirms that $\theta^g$ successfully captures majority/general knowledge, while $\theta^e$ is critical for minority expertise.
>
> - New Visual Demonstration: To visually validate this, we conducted an experiment on Imb. ArtBench-10, IR = 100, where we generated samples for the minority class using only the general branch ($\theta^g$) and compared them with the full model ($\theta^g \oplus \theta^e$). This visual comparison empirically validates that our **Capacity Manipulation** effectively disentangles feature learning: generic features are stored in $\theta^g$, while minority-specific traits are preserved in $\theta^e$. We will add this visualization (Figure G.4) to the Appendix.
>
>
> > My last question is about the diversity within the tail itself. Can you visualize them?
>
> Yes, we have provided extensive visualizations of the diversity within tail (minority) classes in Appendix G.7.
>
> - **Visual Evidence:**
>
>    - **Figure G.6** (Imb. CelebA-HQ, Minority Class: "Male"): Our method generates faces with diverse attributes (e.g., angles, expressions, lighting, and accessories), whereas the baseline tends to produce repetitive patterns.
>    - **Figure G.9 & G.10** (Imb. CelebA & ArtBench-10): Randomly selected samples from few-shot classes demonstrate significant variation in object pose and background, confirming that our model avoids mode collapse in the tail.
> -  **Quantitative Evidence:** This visual diversity is quantitatively supported by our Recall scores (Table 2), a metric specifically designed to measure distributional coverage. Our method achieves the highest Recall on Imb. CIFAR-10/100 and ArtBench-10, confirming superior diversity in the generated tail distributions.
>
> We have added more visualization (Figure G.5) in the revision.

---

### Official Review · Reviewer_pAbs · 2025-11-01

**Soundness:** 3
**Presentation:** 3
**Contribution:** 3
**Rating:** 6
**Confidence:** 3

**Summary:**

This paper addresses the poor minority-class performance of diffusion models trained on long-tailed data, arguing that a key culprit is capacity misallocation—majority classes dominate parameter updates and monopolize representational space.
To tackle this, it proposes Capacity Manipulation (CM): each weight matrix is decomposed into a general/majority component and a reserved low-rank minority component, and training employs a capacity-manipulation loss that enforces consistency for majority classes while promoting diversity for minority classes.
At inference, parameters are merged, introducing no additional latency.
Across imbalanced CIFAR-10/100, CelebA-HQ, ImageNet-LT, iNaturalist, and ArtBench-10 (including Stable Diffusion fine-tuning), CM improves FID/KID and delivers especially strong gains on Medium/Few splits over strong baselines (e.g., CBDM, OC), while remaining orthogonal and complementary to them.

**Strengths:**

The paper offers a clear and original lens—capacity allocation—and introduces a simple, effective mechanism that reserves low-rank capacity for minority classes, moving beyond reweighting or oversampling.
Method quality is strong: the parameter split plus a consistency/diversity loss is minimally invasive, theoretically motivated by gradient/representation analyses, and incurs no inference overhead due to weight merging.
Empirically, results are broad and convincing across multiple datasets/backbones (including SD fine-tuning), with especially large gains on Medium/Few splits and stable ablations over ranks and loss weights.
The approach is practical and orthogonal to existing long-tail remedies (e.g., CBDM, OC), making it easy to adopt and combine for further improvements.

**Weaknesses:**

1. The paper should more clearly distinguish CM from class-balanced objectives, reweighting/oversampling, class-specific adapters/LoRA, and Mixture-of-Experts. Add a side-by-side comparison and reproduce at least one adapter/MoE-style baseline under matched compute.

2. The analysis explains majority gradient dominance and motivates reserving rank, but does not specify conditions ensuring no loss of global likelihood or bounds on interference.

3. Most results are class-conditional image benchmarks.
For instance, text-to-image (multi-attribute, compositional) and multi-label long-tails are underexplored.

**Questions:**

1. How is the minority/majority split determined, and how sensitive are results to this choice under dataset drift or rebalancing?

2. When merging weights at inference, how do the authors prevent cross-talk between the general and minority subspaces?

3. Does reserving capacity degrade majority-class fidelity or diversity in any regimes?

4. What are the exact training overheads introduced by the extra low-rank factors and CM loss? Do gains persist under tight compute budgets?

---

> ### Author Response · Authors · 2025-11-21
> **Response to Reviewer pAbs (1/3)**
>
> > The paper should more clearly distinguish CM from class-balanced objectives, reweighting/oversampling, class-specific adapters/LoRA, and Mixture-of-Experts. Add a side-by-side comparison and reproduce at least one adapter/MoE-style baseline under matched compute.
>
> We appreciate the reviewer's request for clearer positioning. We have added a conceptual comparison and a new experimental comparison with a Class-Specific Adapter baseline.
>
> - **Conceptual Distinctions**: We summarize the differences in the table below. Our CM is unique in that it requires zero inference overhead (unlike MoE/Adapters which often require routing or loading specific weights) and employs a joint training paradigm (unlike standard LoRA fine-tuning) to explicitly reserve capacity via our novel loss.
>
> Comparison of CM with related paradigms.
> | Method | Mechanism | Training Paradigm | Inference Overhead |
> | :--- | :--- | :--- | :--- |
> | Reweighting / RS | Data/Loss Weighting | Single Stage | None |
> | Class-Balanced Obj. (e.g., OC) | Loss Design | Single Stage | None |
> | Adapters / LoRA | Param. Addition | Usually Fine-tuning | None (if merged) / Low |
> | Mixture-of-Experts (MoE) | Dynamic Routing | Joint Training | High (Routing + Experts) |
> | CM (Ours) | Capacity Reservation | Joint Training | Zero (Merged Weights) |
>
> -  **Experimental Comparison (Matched Compute)**:The reviewer requested a comparison with Adapter/MoE-style baselines under matched compute.
>     - **vs. LoRA**: As shown in **Table 1** of our paper, we compared CM against *CBDM (LoRA)*, where we fine-tuned the model using LoRA with the same rank budget. CM outperformed it significantly (FID 7.73 vs 9.73 on CIFAR-10), proving that *how* parameters are optimized (Capacity Manipulation) matters more than just *adding* parameters.
>     - **vs. Class-Specific Adapter (MoE-style):** We implemented a new baseline, **"Group-Expert LoRA"**, which assigns distinct LoRA adapters to "Many", "Medium", and "Few" class groups (acting as a deterministic MoE).
>
> Comparison with MoE-style Baseline on Imb. CIFAR-100 ($\mathrm{IR}=100$).
> | Method | Architecture |  FID $\downarrow$ |
> | :--- | :--- | :--- |
> | DDPM | Standard U-Net |  10.16 |
> | Group-Expert LoRA (MoE-style) | 3 Experts (Many/Med/Few) |  10.06 |
> | CM (Ours) | Capacity Reservation |  7.52 |
>
>
> > The analysis explains majority gradient dominance and motivates reserving rank, but does not specify conditions ensuring no loss of global likelihood or bounds on interference.
>
> We acknowledge the theoretical insight. While we do not derive explicit bounds, we ensure global likelihood and manage interference through our optimization design and verify this empirically.
>
> -  Preservation of Global Likelihood: Our total objective (Eq. 4) explicitly includes the base diffusion loss $\mathcal{L}_{base}(\mathcal{D}, \theta^g \oplus \theta^e)$. This term optimizes the variational lower bound on the log-likelihood for the entire dataset using the merged parameters. This ensures that the global likelihood is maximized jointly, preventing the reserved capacity from detaching from the global data distribution.
>
> - Controlled Interference via Loss Design: Instead of minimizing interference blindly, we structure it. The Capacity Manipulation loss ($\mathcal{L}_{CM}$) explicitly directs the interference:
>    - $\mathcal{L}_{Con}$ forces the general branch $\theta^g$ to align with the full model on majority samples.
>    - $\mathcal{L}_{Div}$ forces the expert branch $\theta^e$ to capture the residual information (minority features) that $\theta^g$ misses.
>
> This "orthogonalization" ensures distinct roles rather than destructive conflict.
>
> - Empirical Validation: Table 3 demonstrates that CM improves minority performance (Few FID: $22.18 \to 18.72$) without degrading majority performance (Many FID: $14.06 \to 11.71$) compared to DDPM on Imb. CIFAR-100. This confirms that allocating capacity to minorities does not come at the cost of global likelihood or destructive interference with majority knowledge.

---

> ### Author Response · Authors · 2025-11-21
> **Response to Reviewer pAbs (2/3)**
>
> > Most results are class-conditional image benchmarks. For instance, text-to-image (multi-attribute, compositional) and multi-label long-tails are underexplored.
>
> We appreciate the reviewer's insight regarding broader application settings. We would like to clarify that we have indeed explored text-to-image generation in our submission.
>
> - Text-to-Image Experiments (Appendix G.8): As detailed in Appendix G.9 and Table G.11, we evaluated CM on fine-grained text-to-image generation using the CUB-200 and Oxford-102 datasets.
>     - **Setup:** We trained Latent Diffusion Models (LDMs) from scratch, using **textual descriptions** as conditioning
>     - **Results:** As shown in Table G.11 (reproduced below), CM consistently outperforms the baseline in these text-conditioned settings, demonstrating its effectiveness beyond simple class labels.
>
> | Dataset | Baseline FID | +CM (Ours) FID |
> | :--- | :---: | :---: |
> | CUB-200 | 24.62 | 22.64 |
> | Oxford-102 | 26.47 | 24.31 |
>
> - Scope regarding Multi-label/Compositional:
>
> While our primary focus is on class-imbalanced distributions (long-tailed recognition), the success on CUB/Oxford (which involves fine-grained attribute descriptions) suggests that our "Capacity Reservation" strategy effectively protects rare concepts (minority text attributes). We agree that multi-label and complex compositional long-tails are exciting avenues for future work, but we believe the current text-to-image results sufficiently demonstrate the method's generalizability.
>
> > How is the minority/majority split determined, and how sensitive are results to this choice under dataset drift or rebalancing?
>
> The split choice is primarily for evaluation analysis and does not affect our method's training or overall performance.
>
> - **Determination of Splits:** Following standard protocols (Jiang et al., 2021), we sort classes by sample frequency and divide them into three equal groups: **Many** (top 1/3), **Medium** (middle 1/3), and **Few**(bottom 1/3)
> - **Robustness in Training:** Our training process is **independent** of these artificial splits. The capacity manipulation weights ($\omega_{Con}$ and $\omega_{Div}$) are calculated using a continuous function of the exact sample count $N_c$ for each class (Eq. 3)2. Therefore, our method adapts dynamically to dataset drift or rebalancing based on actual data distribution, without relying on hard thresholds.
> - **Robustness in Evaluation:** The definition of the split only affects the *reporting* of fine-grained analysis (i.e., Many/Medium/Few FIDs). The **Overall FID** remains unchanged regardless of how the splits are defined.
>
> > When merging weights at inference, how do the authors prevent cross-talk between the general and minority subspaces?
>
> We clarify that the "cross-talk" (or gradient dominance) primarily hinders the learning process, which we resolve during training. The merging at inference is designed to be additive and complementary rather than interfering, based on two key principles:
>
> - Linearity of Layers: Our decomposition applies to linear and convolutional layers. Due to the property $(W^g + W^e)x = W^g x + W^e x$, merging weights at inference is mathematically equivalent to summing their respective feature outputs. There is no dynamic interference mechanism required at inference time.
> - Complementary Optimization Targets: The "prevention" of cross-talk happens during training via our loss design.
>     - **$\mathcal{L}\_{base}$ and $\mathcal{L}\_{Con}$** guide $W^g$ to capture the majority-dominated general distribution.
>     - **$\mathcal{L}\_{Div}$** explicitly forces $W^e$ to learn the **residuals** or specific features that $W^g$fails to capture for minority classes.
>
> Therefore, at inference, $W^e$ acts as a "correction term" that injects the missing minority details into the robust base features provided by $W^g$. This ensures that the merged weight $W$ possesses both general semantic coherence and specific minority fidelity.

---

> ### Author Response · Authors · 2025-11-21
> **Response to Reviewer pAbs (3/3)**
>
> > Does reserving capacity degrade majority-class fidelity or diversity in any regimes?
>
> No, our method does not degrade majority-class performance; in fact, it often improves it.
>
> Our empirical results consistently show improvements or comparable performance on majority classes:
> - Imb. CIFAR-100 (IR=100): As shown in **Table 3**, the FID for the **"Many" (majority)** split improves significantly from **14.07 (DDPM)** to **11.71 (CM)**.
>
> - Imb. CelebA-HQ (IR=100): As shown in **Table 4**, the fidelity of the majority class (**Female**) improves from **7.14** to **6.82**.
>
> Rather than "stealing" capacity, our decomposition ($W = W^g + W^e$) mitigates gradient interference. By isolating minority-specific updates to $W^e$, the general parameters ($W^g$) can capture majority and generalized patterns more stably without being disrupted by conflicting gradients from hard minority samples.
>
> > What are the exact training overheads introduced by the extra low-rank factors and CM loss? Do gains persist under tight compute budgets?
>
> We have analyzed the overheads and evaluated performance under constrained compute budgets. The results show that CM introduces negligible parameter overhead and acceptable training time costs, while maintaining significant advantages even under tight budgets.
>
> - Training Overheads: Calculating the capacity manipulation loss $\mathcal{L}\_{CM}$ requires an additional forward pass to compute the output of the generalized branch ($\epsilon\_{\theta^g}$). This results in a training time increase of approximately **1.3$\times$ to 1.4$\times$** per iteration.
>
> - Performance under Tight Compute Budgets: To demonstrate that our gains stem from efficient capacity allocation rather than increased compute, we compared CM (standard 300k training steps) against strong baselines trained for a significantly longer duration (500k steps).
>
> | Method             | Training Steps | FID ↓ |
> |:------------------ |:--------------:|:-----:|
> | OC (Best Baseline) |      300k      | 8.31  |
> | OC (Extended)      |      500k      | 8.15  |
> | CM (Ours)          |      300k      | 7.52  |

---

### Official Review · Reviewer_7n4d · 2025-11-01

**Soundness:** 3
**Presentation:** 3
**Contribution:** 3
**Rating:** 6
**Confidence:** 3

**Summary:**

This paper proposes Capacity Manipulation (CM), a method to improve diffusion models trained on class-imbalanced data. It identifies that majority classes dominate model capacity, limiting minority representation. CM explicitly reserves capacity for minority classes through low-rank parameter decomposition and a capacity manipulation loss that balances consistency and diversity. Experiments on multiple benchmarks show that CM consistently enhances minority-class generation quality and overall robustness.

**Strengths:**

1. The paper is clearly written, well-structured, and easy to follow.
2. The proposed Capacity Manipulation (CM) method is conceptually simple yet effective, relying on low-rank decomposition and a targeted regularization loss to reserve model capacity for minority expertise.
3. Theoretical analyses provide solid intuition about how imbalance affects parameter updates and how CM mitigates this effect.
4. Extensive experiments across small- and large-scale datasets convincingly demonstrate that CM improves minority-class quality without degrading majority-class performance.

**Weaknesses:**

1. The calculation of loss change in figure 1(b) is not explained.
2. Although the authors evaluate CM across multiple datasets, there is limited discussion of failure cases or sensitivity to extreme imbalance ratios beyond 100:1.
3. Some comparisons (e.g., with Overlap Optimization) are only mentioned in passing, a direct experimental comparison would strengthen claims of superiority.

**Questions:**

N/A

---

> ### Author Response · Authors · 2025-11-21
> **Response to Reviewer 7n4d**
>
> > The calculation of loss change in figure 1(b) is not explained.
>
> We thank the reviewer for pointing this out. We calculate the **Relative Loss Change** to measure the sensitivity of different classes to parameter pruning.
>
> The calculation process is as follows:
>
> 1. **Pruning:** We identify the top 10% of model parameters with the smallest L1-norms and set them to zero.
> 2. **Calculation:** Let $\mathcal{L}\_{raw}^c$ be the original training loss for class $c$, and $\mathcal{L}\_{pruned}^c$ be the loss after pruning. The relative loss change is defined as: $\text{Relative Loss Change}\^c = \frac{\mathcal{L}\_{pruned}^c - \mathcal{L}\_{raw}^c}{\mathcal{L}\_{raw}^c}$
>
>
> This metric highlights that although raw losses are similar across classes2, minority classes are significantly more sensitive to capacity reduction (pruning), suggesting they rely on "fragile" parameters (small L1-norm) rather than the robust core capacity dominated by majority classes.
>
> We have added this explicit definition in the revision.
>
> > Although the authors evaluate CM across multiple datasets, there is limited discussion of failure cases or sensitivity to extreme imbalance ratios beyond 100:1.
>
> We appreciate the reviewer’s suggestion to clarify the method's boundaries. We address the robustness to extreme imbalance and discuss potential failure cases below:
>
> - Robustness to Extreme Imbalance (IR > 100): We respectfully highlight that our paper includes evaluations on large-scale datasets with extreme imbalance ratios significantly beyond 100:1:
>     - **ImageNet-LT:** Imbalance Ratio (IR) = **256**
>     - **iNaturalist:** Imbalance Ratio (IR) = **500**
> As shown in **Table 5**, CM consistently outperforms baselines on these datasets. Specifically, on iNaturalist ($64\times64$), CM reduces FID by **3.46** compared to CBDM and **0.85** compared to OC4, demonstrating that our capacity reservation strategy remains effective even under extreme long-tailed distributions.
>
> - **Discussion of Failure Cases (Absolute Scarcity)**: While CM effectively mitigates the dominance of majority classes, we have identified specific scenarios where the method faces limitations:
>     - **Absolute Sample Scarcity:** When the *absolute* number of minority samples is extremely low (e.g., single-digit samples in the tail of iNaturalist), the reserved capacity $\theta^e$ lacks sufficient data to learn meaningful representations, regardless of the allocated parameter space. In these "few-shot" to "zero-shot" transitions, the generation quality remains limited.
>
> We have added these discusions in the revision to discuss these boundaries explicitly (Appendix I).
>
> > Some comparisons (e.g., with Overlap Optimization) are only mentioned in passing, a direct experimental comparison would strengthen claims of superiority.
>
> We appreciate the reviewer's suggestion to strengthen our baselines. We have now conducted a direct experimental comparison with Overlap Optimization (Yan et al., 2024) on the Imb. CIFAR-100 ($IR=100$) dataset.
>
> As shown in the table below, our method (CM) outperforms Overlap Optimization across all key metrics.
>
> Comparison with Overlap Optimization on Imb. CIFAR-100 (IR=100).
>
> | Method | FID $\downarrow$ | Recall $\uparrow$ |
> | :--- | :---: | :---: |
> | DDPM | 10.16 | 0.46 |
> | Overlap Opt. (Yan et al., 2024) | 8.98 | 0.48 |
> | CM (Ours) | 7.52 | 0.52 |

---

### Official Review · Reviewer_NJJB · 2025-11-06

**Soundness:** 3
**Presentation:** 3
**Contribution:** 3
**Rating:** 6
**Confidence:** 4

**Summary:**

This paper addresses the problem of class imbalance in diffusion models, which leads to poor generation performance on minority classes. The authors identify model capacity allocation as a key overlooked factor, where majority classes dominate model parameters, leaving insufficient capacity for minorities. To mitigate this, they propose Capacity Manipulation (CM), a method that reserves model capacity for minority classes via low-rank decomposition of parameters and a novel capacity manipulation loss. The method is orthogonal to existing approaches and does not increase inference cost. Extensive experiments on multiple datasets demonstrate consistent improvements in minority-class generation without sacrificing majority-class performance.

**Strengths:**

(1) The method is well-motivated, supported by both empirical observations (e.g., pruning sensitivity) and theoretical analysis (Theorems 2.1 & 3.1). The experimental setup is rigorous, covering multiple datasets, architectures, and metrics.

(2) The paper offers an orthogonal viewpoint on class imbalance in diffusion models by focusing on model capacity allocation, diverging from prior works that primarily emphasize loss reweighting or knowledge transfer (e.g., CBDM and OC). The integration of low-rank decomposition with a tailored loss function represents a creative combination of ideas from parameter-efficient fine-tuning and imbalanced learning for targeted capacity reservation.

**Weaknesses:**

(1) The term "capacity" is not clearly defined. Is it the number of parameters, the magnitude (e.g., L1-norm) of the weights or something else? The pruning experiment suggests a link to weight magnitude, but this connection is not explicitly made or theoretically grounded. Therefore, " capacity" remains a somewhat vague concept.

(2) The capacity manipulation loss is designed to force minority-specific knowledge into the low-rank adapter. A potential risk is that this adapter becomes too specialized, failing to leverage the shared, general features learned by the main model. This could limit its ability to generate diverse minority samples that still rely on common underlying features (e.g., a "rare breed of dog" should still benefit from general "dog" features). The paper does not discuss or analyze this potential limitation.

**Questions:**

(1) The method proposed in the paper primarily focuses on the context of known classes. A natural follow-up question is how Capacity Manipulation would perform in scenarios involving more compositional and fine-grained concepts. For example, in a dataset imbalanced towards "photos of cats" vs. "paintings of dogs," how would the model reserve capacity for the minority concept of "painting" style, which is orthogonal to the object "dog"? Does this framework extend to reserving capacity for concepts rather than just classes?

(2) The method's architecture—using a LoRA-like adapter—makes a strong, implicit assumption: that "minority expertise" is inherently low-rank. What is the theoretical or empirical justification for this? One could easily argue the opposite: minority classes might be more complex and have a higher intrinsic dimensionality (e.g., "impressionist painting" vs. "female face") but are simply under-sampled. If the minority knowledge is, in fact, high-rank, then the fixed low-rank of the adapter would become the primary performance bottleneck, ironically limiting the minority class's capacity more than a standard full-rank model. How does CM cope with this potential issue?

(3) The current formulation appears to use a single $\theta^e$ to capture the expertise for all minority classes collectively. On datasets with highly heterogeneous minority classes (e.g., the "Few" split in Imb. CIFAR-100 or ImageNet-LT, which can contain wildly different concepts), is it plausible that a single low-rank subspace can effectively represent this diverse and multimodal knowledge? Does this not create a new "capacity collapse" problem within the minority adapter itself? Have the authors considered a more flexible architecture, such as a Mixture-of-Experts (MoE) model for $\theta^e$, where different "experts" (adapters) are dynamically allocated to different minority clusters?

(4) The paper should discuss and cite relevant literature on reweighting or balancing techniques for generative models [1-5].

Reference:

[1] Xie et al. Doremi: Optimizing data mixtures speeds up language model pretraining. NeurIPS, 2023.

[2] Fan et al. DoGE: Domain Reweighting with Generalization Estimation. ICML, 2024.

[3] Kim et al. Training unbiased diffusion models from biased datase. ICLR, 2024.

[4] Li et al. Pruning then Reweighting: Towards Data-Efficient Training of Diffusion Models. ICASSP, 2025.

[5] Liu et al. RegMix: Data Mixture as Regression for Language Model Pre-training. ICLR, 2025.

---

> ### Author Response · Authors · 2025-11-21
> **Response to Reviewer NJJB (1/3)**
>
> number of parameters, the magnitude (e.g., L1-norm) of the weights or something else? The pruning experiment suggests a link to weight magnitude, but this connection is not explicitly made or theoretically grounded. Therefore, " capacity" remains a somewhat vague concept.
>
> We thank the reviewer for pointing out the ambiguity regarding the term "capacity." We agree that a precise definition is crucial for the clarity.
>
> - Formal Definition of Capacity in Our Context: In this paper, we define "model capacity" as the **available representational resources** of the neural network to capture and store data features.  We will add a formal definition of "Model Capacity" in the revision. Specifically, we operationalize this concept in two complementary ways:
>     - **Structural Perspective (Rank):** Following the principles of Low-Rank Adaptation (LoRA), we treat the rank of the weight matrices as a direct measure of capacity. A full-rank matrix has maximum capacity, while a low-rank decomposition limits the information flow. Our method ($W = W^g + W^e$) explicitly manipulates this by reserving a specific rank ($r$) for minority expertise.
>     - **Optimization Perspective (Gradient Dominance):** Theoretically, we define "capacity allocation" based on which classes dominate the parameter updates. As proven in Theorem 2.1, majority classes contribute significantly larger aggregated gradients, thereby dominating the update direction ($\Delta W$) and effectively "monopolizing" the parameters.
> - Clarification on the Pruning Experiment (L1-norm). We wish to clarify that the **L1-norm magnitude is a diagnostic indicator (proxy), not the definition itself.**
>     - In deep learning, parameters with larger magnitudes typically contribute more to the model's output and loss reduction. Our pruning experiment (Fig. 1b) reveals that minority classes are disproportionately sensitive to the removal of *small* magnitude parameters. This implies that majority classes occupy the "strong" connections (high capacity usage), forcing minority knowledge to be encoded in "weak" connections (marginal capacity). This observation empirically supports our motivation to explicitly reserve capacity via $\theta^e$.
>
>
> >  The capacity manipulation loss is designed to force minority-specific knowledge into the low-rank adapter. A potential risk is that this adapter becomes too specialized, failing to leverage the shared, general features learned by the main model. This could limit its ability to generate diverse minority samples that still rely on common underlying features (e.g., a "rare breed of dog" should still benefit from general "dog" features). The paper does not discuss or analyze this potential limitation.
>
> Thank you for highlighting this important point. We clarify that our design specifically mitigates the risk of the adapter becoming an isolated "island" through two key mechanisms:
>
> - **Additive Architecture & Joint Training:** As defined in Eq. (1), the final model parameters are the sum of the general and specific components: $\theta = \theta^g \oplus \theta^e$. Crucially, we optimize the **Base Loss ($\mathcal{L}_{base}$)** on the **combined parameters** across the **entire dataset** (including minorities).
>     - This ensures that $\theta^g$ learns the robust, high-frequency general features (e.g., the structure of a "dog") shared by all classes.
>     - $\theta^e$ functions as a **residual correction** rather than a standalone generator. Because $\theta^e$ is low-rank3, it lacks the capacity to generate high-fidelity images on its own; it *must* rely on the structural foundation provided by $\theta^g$.
> - **Evidence of Feature Sharing (Ablation Study):** Our ablation study in **Table 7** supports this cooperative behavior
>     - **CM ($\theta^g$ only):** Performs well on "Many" classes (general features) but fails on "Few" classes (lacking specific details).
>     - **CM (Full):** drastically improves "Few" class performance while maintaining "Many" class quality.
>     - This confirms that minority generation relies on the *combination*: utilizing $\theta^g$ for structure (common features) and $\theta^e$ for specific attributes (rare breed details).

---

> ### Author Response · Authors · 2025-11-21
> **Response to Reviewer NJJB (2/3)**
>
> > The method proposed in the paper primarily focuses on the context of known classes. A natural follow-up question is how Capacity Manipulation would perform in scenarios involving more compositional and fine-grained concepts. For example, in a dataset imbalanced towards "photos of cats" vs. "paintings of dogs," how would the model reserve capacity for the minority concept of "painting" style, which is orthogonal to the object "dog"? Does this framework extend to reserving capacity for concepts rather than just classes?
>
> We thank the reviewer for this insightful question regarding compositional concepts. We believe our Capacity Manipulation (CM) framework is well-suited for such scenarios, extending effectively to reserving capacity for specific concepts (like "style") implicit in the minority labels.
>
> - **Mechanism for Capturing Orthogonal Concepts**: While our method uses class labels for supervision, the mechanism of CM naturally disentangles concepts based on frequency.
>     - Recall that we decompose parameters into $W^g$ (General/Majority) and $W^e$ (Minority Expertise)
>     - In the "photo of cats" (majority) vs. "painting of dogs" (minority) example, $W^g$ is trained to capture the dominant data distribution (e.g., photorealistic textures, general object structures).
>     - The diversity loss $\mathcal{L}_{Div}$ explicitly forces the reserved capacity $W^e$ to capture the *residual* information necessary to reconstruct the minority samples—features that $W^g$ fails to represent. Therefore, if "painting style" is the key orthogonal feature distinguishing the minority data, $W^e$ will be optimized to encode this specific stylistic concept.
>
> -  **Empirical Evidence on Fine-Grained Concept**s:
>     - We also evaluated CM on fine-grained datasets like CUB-200 and Oxford-102 (Appendix G.9), where class differences rely on subtle visual attributes (concepts) rather than distinct object categories. CM consistently improved FID scores, further validating its ability to manage fine-grained compositional concepts.
>
> > The method's architecture—using a LoRA-like adapter—makes a strong, implicit assumption: that "minority expertise" is inherently low-rank. What is the theoretical or empirical justification for this? One could easily argue the opposite: minority classes might be more complex and have a higher intrinsic dimensionality (e.g., "impressionist painting" vs. "female face") but are simply under-sampled. If the minority knowledge is, in fact, high-rank, then the fixed low-rank of the adapter would become the primary performance bottleneck, ironically limiting the minority class's capacity more than a standard full-rank model. How does CM cope with this potential issue?
>
>
> We thank the reviewer for this insightful observation regarding intrinsic dimensionality. We would like to clarify that our method **does not assume** minority knowledge is inherently low-rank. Rather, our motivation stems from an **optimization perspective** concerning sample scarcity and gradient dominance.
>
> - **Protected Capacity vs. Dominated Capacity**: In standard full-rank training, parameters are shared globally. As shown in Theorem 2.1 and Fig. 1(b), majority classes dominate the gradient updates, causing the effective capacity available to minority classes to collapse near zero. Therefore, the trade-off is not between "low-rank" and "full-rank" capacity for the minority, but rather between:
>     - **Standard:** A full-rank space where minority gradients are overwhelmed and overwritten by majority updates.
>     - **Ours (CM):** A **protected** low-rank space ($W^e$) that is explicitly reserved for minority signals and shielded from majority encroachment via our Capacity Manipulation loss.
> - **Sufficient Capacity for Sparse Samples**: While minority concepts may be complex, the available information is limited due to the low sample count. A protected low-rank adapter is sufficient to capture the learnable features present in the limited data without overfitting, whereas a shared full-rank model fails to retain them entirely.
>
> In short, we provide a **"safe harbor"** for minority features: a dedicated low-rank space is far superior to a full-rank space that the minority class effectively cannot access.

---

> ### Author Response · Authors · 2025-11-21
> **Response to Reviewer NJJB (3/3)**
>
> > The current formulation appears to use a single  to capture the expertise for all minority classes collectively. On datasets with highly heterogeneous minority classes (e.g., the "Few" split in Imb. CIFAR-100 or ImageNet-LT, which can contain wildly different concepts), is it plausible that a single low-rank subspace can effectively represent this diverse and multimodal knowledge? Does this not create a new "capacity collapse" problem within the minority adapter itself? Have the authors considered a more flexible architecture, such as a Mixture-of-Experts (MoE) model for , where different "experts" (adapters) are dynamically allocated to different minority clusters?
>
> We thank the reviewer for this insightful suggestion. We agree that a Mixture-of-Experts (MoE) design is a promising direction to further scale our framework. However, we believe our current single-adapter design is justified and effective for the following reasons:
>
> - **Empirical Effectiveness on Heterogeneous Data**: Our experiments on ImageNet-LT (1,000 classes) and iNaturalist (8,142 classes) involve highly heterogeneous minority concepts. As shown in Table 5, CM significantly outperforms baselines (e.g., improving FID from 12.61 to 10.94 on ImageNet-LT). This empirically demonstrates that a single low-rank $\theta^e$ does not suffer from "capacity collapse" but effectively safeguards minority knowledge.
> - **Role of $\theta^e$ vs. $\theta^g$**: The primary goal of $\theta^e$ is not to learn all features of minority classes from scratch, but to provide a dedicated subspace protected from the gradient dominance of majority classes (as analyzed in Theorem 3.1). The shared backbone $\theta^g$ still captures generalized semantic features applicable to all classes3. Thus, $\theta^e$ mainly acts as a robust "correction" or "protection" mechanism, which requires less capacity than learning full representations.
> - **Efficiency and Simplicity**: A key advantage of our current design is its zero inference overhead. By merging $\theta^g$ and $\theta^e$ after training ($\theta = \theta^g \oplus \theta^e$), we maintain the standard inference speed. While an MoE architecture could potentially offer finer-grained capacity allocation, it would introduce complexity in routing and potentially increase training/inference costs.
>
> We regard our current formulation as a foundational step that verifies the critical role of capacity manipulation. We explicitly acknowledge the MoE-based adapter as a valuable extension for future work to handle even more complex distributions.
>
> > The paper should discuss and cite relevant literature on reweighting or balancing techniques for generative models [1-5].
>
> We appreciate the suggestion. We have added the relevant citations [1-5] and briefly discussed these reweighting/balancing techniques in the revised manuscript to provide a more comprehensive background (Appendix B).

---

### Author Response · Authors · 2025-11-21
**General Response**

We thank the reviewers for their time, constructive feedback, and encouraging assessment of our work. We are gratified that the reviewers reached a consensus on the novelty and effectiveness of our approach. Specifically, we appreciate that the reviewers recognized:

- **Novel & Orthogonal Perspective:** Reviewers praised our **identification of "model capacity allocation"** as a key factor in class imbalance, noting it as a "**fresh perspective**" (JwQN) and an "**original lens**" (pAbs) that is "**orthogonal**" to existing loss-reweighting approaches (NJJB).
- **Strong Theoretical Foundation:** Reviewers highlighted that our method is "**well-motivated**" (NJJB) and supported by "**solid intuition**" (7n4d) and theoretical analysis (Theorems 2.1 & 3.1) that substantiates the **core thesis** of majority gradient dominance (JwQN).
- **Effectiveness & Efficiency:** Reviewers commended the method for being "**conceptually simple yet effective**" (7n4d), yielding "**comprehensive**" experimental validation (JwQN) with "**strong gains**" on tail classes (pAbs), all while introducing "**no additional inference overhead**" (NJJB, pAbs).

In response to the valuable suggestions, we have extensively revised the paper (changes highlighted in **teal**). A summary of the key updates follows:

**1. New Baselines and Comparative Experiments**

- **Comparison with MoE-style Baseline:** To distinguish our method from parameter-efficient fine-tuning or routing techniques, we implemented a "Group-Expert LoRA" baseline. Results confirm that our explicit capacity reservation significantly outperforms simple expert assignment (pAbs).
- **Comparison with Overlap Optimization:** We added a direct comparison with the concurrent work *Overlap Optimization* (Yan et al., 2024), demonstrating that CM achieves superior FID and Recall (7n4d).

**2. Extension to Broader Tasks**

- **Discriminative Tasks:** We extended CM to long-tailed recognition (CIFAR-100 with ResNet-18). Experiments show that CM consistently improves strong baselines (LDAM, Logit Adjustment), validating the generalizability of the "capacity reservation" principle beyond generative models (JwQN).
- **Toy Experiment:** We added a 2D toy experiment visualizing gradient directions, empirically confirming Theorem 2.1 that majority classes dominate optimization trajectories (JwQN).

**3. Clarifications and Visualizations**

- **Definition of Capacity:** We added a formal definition of "Model Capacity" in Sec. E.1, operationalizing it via both structural rank and optimization/gradient dominance (NJJB).
- **Visual Ablation:** We added visualizations (Fig. G.4) comparing samples from the general branch ($\theta^g$) vs. the full model, empirically proving that generic features are stored in $\theta^g$ while minority-specific traits are preserved in $\theta^e$ (JwQN, NJJB).
- **Metric Definition:** We explicitly defined the "Relative Loss Change" metric used in Fig. 1(b) (7n4d).

**4. Discussion and Literature**

- **Conceptual Comparison:** We added Table E.1 to clearly distinguish CM from reweighting, adapters, and MoE approaches (pAbs).
- **Limitations:** We expanded the discussion on limitations, specifically regarding absolute sample scarcity (few-shot to zero-shot transition) (7n4d).
- **Literature:** We incorporated suggested citations regarding reweighting and data mixture techniques (NJJB).

We believe these revisions significantly strengthen the paper and address the reviewers' questions. We have responded to each reviewer individually below.

Best regards,

Authors of Submission 2255

---

### Meta-Review · Area_Chair_m5NP · 2025-12-29

**Summary:**

This paper addresses class-imbalanced training for diffusion models from a capacity-manipulation perspective. The method is based on low-rank decomposition, where consistent and diversity losses are applied to different sets of model parameters with class-specific weights.

Reviewers commented positively on the well-motivated and well-designed method, rigorous and convincing experiments, a novel perspective on class imbalance based on model capacity, an interesting method combining parameter-efficient fine-tuning and imbalanced learning, and a theoretical analysis.

Reviewers raised questions about the lack of clarity on “capacity”, potential limitations of the method, failure cases, and the need for additional experiments. The concerns have been well addressed.

Overall, reviewers have reached a consensus that this paper is a high-quality contribution.

**Reviewer Concerns:**

(Weaknesses are indexed using reviewers' original ordering)

For Reviewer NJJB, W1 and W2 have been addressed.

For Reviewer 7n4d, W1, W2, and W3 have all been addressed.

For Reviewer pAbs, W1, W2, and W3 have all been addressed.

For Reviewer JwQN, W1-6 have all been addressed.

**Reviewer Scores:**

For Reviewer NJJB, the score is likely to be the same or increased.

For Reviewer 7n4d, the score is likely to be the same or increased.

For Reviewer pAbs, the score is likely to be the same or increased.

For Reviewer JwQN, the score is likely to be the same or increased.

---

### Decision · Program_Chairs · 2026-01-26

Accept (Oral)